# Chameleon-inspired tunable multi-layered infrared-modulating system via stretchable liquid metal microdroplets in elastomer film

Yingyue Zhang[1,2], Hanrui Zhu[1,2], Shun An [1,2], Wenkui Xing [1,2], Benwei Fu[1,3], Peng Tao [1,3], Wen Shang [1,2], Jianbo Wu [1,2], Michael D. Dickey [4] ✉, Chengyi Song [1,2,3] ✉ & Tao Deng [1,2,3] ✉

This report presents liquid metal-based infrared-modulating materials and systems with multiple modes to regulate the infrared reflection. Inspired by the brightness adjustment in chameleon skin, shape-morphing liquid metal droplets in silicone elastomer (Ecoflex) matrix are used to resemble the dispersed "melanophores". In the system, Ecoflex acts as hormone to drive the deformation of liquid metal droplets. Both total and specular reflectance-based infrared camouflage are achieved. Typically, the total and specular reflectances show change of ~44.8% and 61.2%, respectively, which are among the highest values reported for infrared camouflage. Programmable infrared encoding/decoding is explored by adjusting the concentration of liquid metal and applying areal strains. By introducing alloys with different melting points, temperature-dependent infrared painting/writing can be achieved. Furthermore, the multi-layered structure of infrared-modulating system is designed, where the liquid metal-based infrared modulating materials are integrated with an evaporated metallic film for enhanced performance of such system.

Dynamically regulating mid-infrared radiation (mid-IR: 7.5–14 μm) in response to external stimuli enables advanced technologies encompassing encryption[1,2], electrochromic displays[3,4], camouflage[5–7], radiative cooling[8], human-machine interaction[9] and thermal management[10,11]. Specifically, mechanical stimulated IR regulators can be also used in the fields of personal thermal management[10], radiative heat management[12], finger motion sensing[13], and etc. Various materials used in existing dynamic IR modulating-systems can help control certain IR properties of the systems. For example, metals exhibit high IR reflectance[6,7,10], phase-changing materials possess temperature-regulating IR emission[14,15], and two-dimensional (2D) nanomaterials show component-dependent IR emissivity[4]. In addition, most IR-modulating materials reported are designed as 2D structures, which

restricts the development of IR modulation along the out-of-plane direction[4,6,7,10,16–18]. To enable a broad range of practical applications, the ideal IR-modulating materials should not only exhibit superior modulation performance and have multiple mechanisms to regulate the response, but also allow multi-layered design.

In nature, melanophores in chameleon's skin have been discovered to strongly modulate the interaction between incident sunlight and xanthophores/iridophores through the dispersion and aggregation of melanosomes[19,20] (Fig. 1a). When melanosomes are aggregated in a perinuclear position within melanophores, the skin appears brightly colored. In contrast, when the melanosomes migrate into the terminal processes that lie above iridophores and xanthophores, they effectively prevent light from striking the iridophores

[1]The State Key Laboratory of Metal Matrix Composites, School of Materials Science and Engineering, Shanghai Jiao Tong University, 800 Dong Chuan Road, Shanghai 200240, PR China. [2]Center of Hydrogen Science, Shanghai Jiao Tong University, 800 Dong Chuan Road, Shanghai 200240, PR China. [3]National Engineering Research Center of Special Equipment and Power System for Ship and Marine Engineering, 10 Heng Shan Road, Shanghai 200030, PR China. [4]Department of Chemical and Biomolecular Engineering, North Carolina State University, 911 Partners Way, Raleigh, NC 27695, USA. ✉e-mail: mddickey@ncsu.edu; chengyi2013@sjtu.edu.cn; dengtao@sjtu.edu.cn

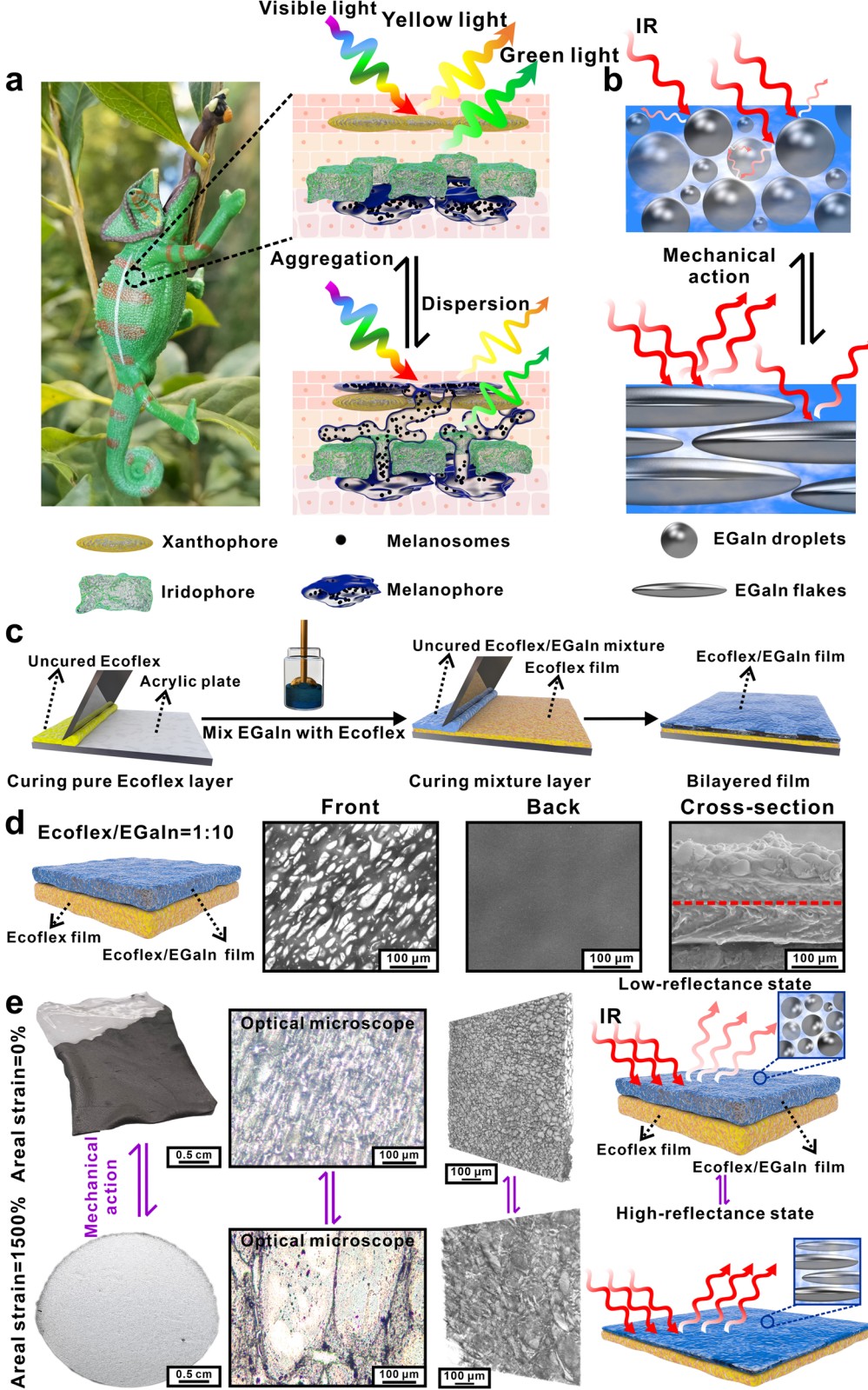

**Fig. 1 | Concept and bio-inspired design of bilayered Ecoflex/EGaIn film (BLEE) enabling IR modulation. a** Schematics of the mechanism of altering the brightness of a chameleon's skin through the expansion/shrink of melanophores in the superficial iridophores of chameleons. **b** Schematic illustration of BLEE film modulating the IR reflectance by applying mechanical action. **c** Schematic illustration of the fabrication process of BLEE film. **d** Scanning electron microscope (SEM) images of the front, back, and cross-section of BLEE film with Ecoflex/EGaIn mass ratio of 1:10. The pure Ecoflex layer was under the red dashed line and the Ecoflex/EGaIn layer was above the red dashed line. **e** Optical appearance of BLEE film with Ecoflex/EGaIn mass ratio of 1:10 before and after applying areal strain of 1500%, images from optical microscopy and X-ray microscope (XRM) of the morphology change of EGaIn droplets before and after stretching.

below. It causes the skin to appear dark. Such a process is fully reversible. The brightness change of chameleon skin involves tuning the interaction between sunlight and iridophores by moving the melanosomes, which would not change the wavelength of reflected light but change the intensity of reflected light. It is similar to the working principle of an IR camera that uses fake color to represent the light intensity of IR ranging from 7.5 to 14 μm. Due to the tunable IR reflectance of the IR modulating system, different IR signals could be regulated and result in an adaptable IR image. Therefore, the brightness change mechanism of a chameleon inspires the design of a reversible IR-modulating system.

Liquid metal microdroplets with reversible shape-morphing properties can act as a candidate for the "melanophores" of the IR modulating system. Featuring water-like viscosity, high surface tension, and low biological toxicity, liquid metals have been widely used in soft circuits[21,22], thermal management[23,24], imaging contrast agents[25,26], and chemical catalysis[27,28]. Liquid metals resemble aluminum (Al) or gold (Au) with strong IR reflectance, yet unlike those solid metals, liquid metal can easily change shape. Moreover, the IR reflectance changes when the metal goes through a phase transition; the melting point (mp) can be tuned by varying the allowed composition. All of these merits allow liquid metals to modulate IR reflectivity.

In this work, a bio-inspired multi-layered structure composed of eutectic gallium-indium (EGaIn) droplets dispersed in a film of silicone elastomer (Ecoflex) has been demonstrated, which enables IR modulation by tuning IR reflectance. The encapsulated EGaIn droplets are used as "melanophores" for IR modulation, which can adjust reflectance by changing their shape from spherical to flakes in response to strain (Fig. 1b). Ecoflex matrix, serving as "hormone", drives the deformation of EGaIn droplets under mechanical strain. Before stretching, the scattering of the spherical EGaIn droplets and IR absorption by the Ecoflex matrix result in a low-IR reflectance state. After stretching, overlapping EGaIn flakes have a metallic luster, leading to a state of high-IR reflectance. Total reflectance and specular reflectance-based IR camouflage have been both achieved in this work. In previous work, a wrinkled surface of polymer was required to regulate specular IR reflectance[7,18]. In this work, the deformation of EGaIn droplets plays a major role in spectral modulation. Using only strain, the specular reflectance changes by 61.2% and apparent temperature changes by nearly 21.2 °C, which is among the highest values of specular reflectance-based IR camouflage[7,18]. Programmable IR encoding/decoding can be achieved by varying the strain and the concentration of liquid metal. Moreover, different alloys have been introduced to change the mp, thereby enabling distinct IR reflectance between solid-and liquid-state metals. We demonstrate applications of these materials for IR display and communication, which is distinct from the 2D pixelated driver array approach[1,4,6,7,13]. The findings not only serve as potential programmable IR camouflage, IR encryption, and IR painting/writing, but also open up a new path for the multiple modes of IR regulation and multi-layered design of materials for applications including IR sensors[29,30], actuators[31–33], flexible electronics[34–37], artificial intelligence[2,9], and thermal management[10,23,24,38].

## Results

### Structural and morphological characterization

BLEE film was fabricated by directly mixing Ecoflex and EGaIn in a specific mass ratio, and coating it onto a film of Ecoflex (Fig. 1c, see the Method for details). The bottom layer (pure Ecoflex) both prevents the leakage of liquid metal and achieves different IR response from the top layer (Ecoflex/EGaIn mixture) since the bottom layer shows constant low IR reflectance and the upper layer exhibits tunable reflectance under external stimuli. The SEM images and the size distributions of liquid metal droplets from composites with different mass ratios are shown in Supplementary Fig. 1. The size distribution of liquid metal

droplets ranges from a few micrometers to tens of micrometers. As for the liquid metal microdroplets mix with polymer at mass ratio of 1:2, the average diameter is 17.94 μm. By increasing the liquid metal concentration in the composites, the size of individual microdroplets decreases. Mixing liquid metal microdroplets with polymer at mass ratio of 1:10 results in the smallest average size of 6.68 μm. It is attributed that the viscosity of the mixture increases with the increasing amount of liquid metal in the composite. The enhanced shear stress during mixing leads to the breakup of the liquid metal droplets and smaller average size distributions[39,40]. To examine the microstructure of as-prepared samples with different Ecoflex/EGaIn mass ratios, SEM has been employed and shown in Fig. 1d, Supplementary Figs. 2 and 3. Both single-layered Ecoflex/EGaIn film and BLEE film contain randomly dispersed EGaIn microdroplets, and the dispersion density of droplets increases with the increase of mass ratios (Supplementary Fig. 2). The morphology changes of stretched Ecoflex/EGaIn film have been examined by optical microscopy. As shown in Fig. 1e and Supplementary Fig. 4, with the increase of areal strain, the appearance of BLEE film changes from a dark gray color to a bright silvery-white color with a metallic luster. It is attributed that EGaIn droplets inside polymer transform from the spherical state into the flake state under mechanical stretching force (Fig. 1e and Supplementary Fig. 5). From the XRM results in Fig. 1e, Supplementary Fig. 6, Supplementary Movies 1–4, the 3D reconstructor analysis shows that the liquid metal droplets are randomly dispersed in the elastomer. After applying areal strain of 1500%, the liquid metal droplets in the composite transform into flakes. High concentration of liquid metals in elastomer (mass ratio of 1:10) create a quasi-continuous layer due to the overlap of liquid metal flakes. And inspired by the brightness adjustment of chameleon skin, the EGaIn droplets can act as a kind of IR melanophore to adjust the IR reflectance signal of the object's surface (various false colors in IR camera). There are two types of IR camouflage: (1) total reflectance-based; and (2) specular reflectance-based IR camouflage. In the former type, reflection of the IR signals from the surrounding environment with relatively low temperatures can cloak underlying objects with high temperatures[4,6,13,41]. In the second type of IR camouflage, cloaking the low-temperature objects can be achieved by specularly reflecting the IR signal emitted from higher-temperature objects[6,7,18].

### Total reflectance-based IR camouflage

We evaluated the IR camouflage performance (total IR reflectance) of BLEE films with defined Ecoflex/EGaIn mass ratios by placing the BLEE on a hot plate to mimic a high-temperature object (Fig. 2a). By varying the Ecoflex/EGaIn mass ratio (1:0, 1:2, 1:4, 1:6, 1:8, and 1:10) and areal strain (0% to 1500%), it is possible to tune the IR reflectance. Figure 2b, c depict the relationship between apparent temperature and areal strain. With the increase of areal strain and concentration of EGaIn, the apparent temperature decreases. Therefore, BLEE films in this work can achieve almost 44.8% of total reflectance change (from 29.8% to 74.6%), and reduce the apparent temperature of a hot object (76 °C) by 34.5 °C. Closely examining the optical images in Supplementary Fig. 5, it is attributed that the increased filling volume of EGaIn would fill in the gaps between the EGaIn droplets after stretching the film, and the expanded EGaIn droplets could form a smooth and continuous shiny metallic surface to reflect incident IR. The apparent temperatures of BLEE films with the areal strain of 1500% have been demonstrated to be linearly proportional to the temperature of the hot plate (Supplementary Fig. 7a, b), indicating that the IR reflectance of a specific BLEE film always maintains the same with the change of object's temperature.

The IR spectral measurement reveals how the deformation of EGaIn droplets impacts the total reflectance in the IR range. In Fig. 2d and Supplementary Fig. 8, the total reflectance of BLEE films with different mass ratios increases with the enlarged areal strain. As

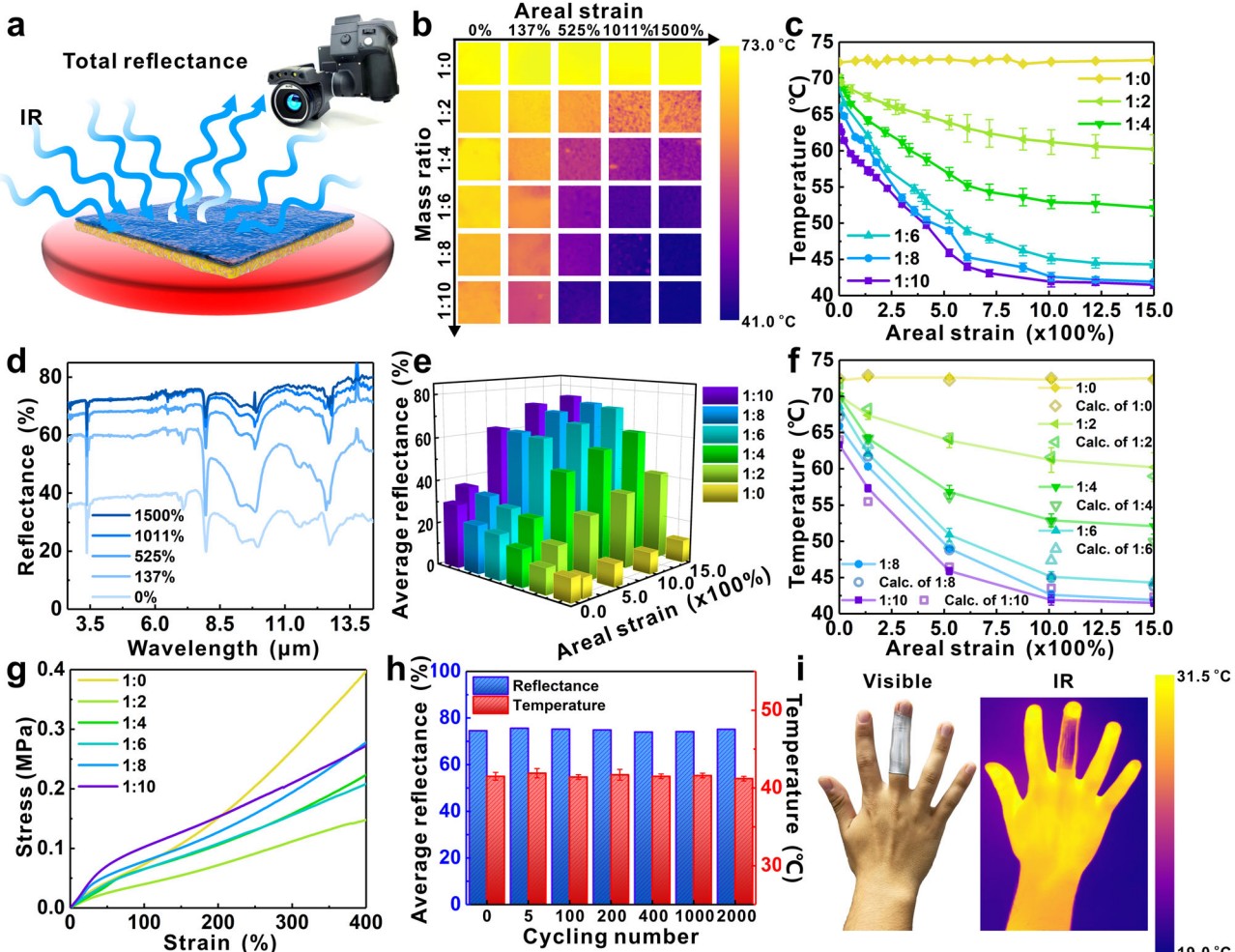

**Fig. 2 | Total reflectance-based IR camouflage performance, reflectance spectra, and mechanical properties of BLEE films. a** Schematic illustration of experimental observation of IR camouflage, where samples are placed on the hot plate and IR camera is used to take pictures. **b** IR images of BLEE films with specific mass ratio of Ecoflex/EGaIn (1:0, 1:2, 1:4, 1:6, 1:8, or 1:10) after applying various areal strains of 0%, 137%, 525%, 1011% and 1500%, respectively. **c** The temperature curves taken from the IR images of BLEE films in Fig. 2b. All error bars represent the standard deviation. **d** The total reflectance spectra of BLEE film with mass ratio of 1:10, after applying different areal strains (0%, 137%, 525%, 1011%, and 1500%). **e** The calculated average reflectances ranging from 7.5 μm to 14 μm. **f** Comparison of apparent temperatures of BLEE films with different mass ratios read from measured IR images (lines) and the calculated apparent temperatures from measured total reflectance spectra (dots). All error bars represent the standard deviation. **g** The measured stress-strain curves of pure Ecoflex and BLEE films with different mass ratios of Ecoflex/EGaIn (1:2, 1:4, 1:6, 1:8, and 1:10) in the range of strain from 0% to 400%. **h** The average reflectances calculated from total reflectance spectra and measured apparent temperatures of BLEE film with mass ratio of 1:10 placed on the hot plate (76 °C) after different cycles (0, 5, 100, 200, 400, 1000, and 2000). All error bars represent the standard deviation. **i** The IR and visible optical photos of stretched BLEE film with mass ratio of 1:10 wrapped human hand to show IR camouflage effect.

confirmed by Eq. (S8), the total reflectance of BLEE film plays a critical role in IR camouflage, since the apparent temperature of BLEE film observed from the IR camera decreases with the increase of total IR reflectance. To quantify the total reflectance detected by IR camera, the average total reflectance has been calculated from the measured spectra (see the calculation details in the Supplementary Materials). The numerical calculation results show that the deformation and the mass ratio of EGaIn droplets are the two key points for controlling the average reflectance (Fig. 2e). Theoretically, calculated apparent temperatures of stretched BLEE films in terms of the average reflectance (Fig. 2f, Supplementary Tables 1 and 2) (see the calculation details in the Supplementary Materials) are in good agreement with experimentally measured values (Fig. 2c), which further confirm the relationship between total reflectance and apparent temperature. Well-designed droplet size of liquid metal is also important in the BLEE-based IR-modulating materials. We fabricated BLEE films with different liquid metal droplet size distributions and measured their IR

reflectance in Supplementary Figs. 9 and 10. BLEE films composed of hundred-nanometer liquid metal droplets could not show mechanically stimulated IR modulation, since the liquid metal nanodroplets cannot undergo deformation in the polymer matrix[42]. We also performed theoretical simulations and confirmed this size-dependent deformation of liquid metal droplets in Ecoflex matrix (see the Note in Supplementary Materials and Supplementary Fig. 11), which has been also observed in previous work[43].

The mechanical properties of BLEE films have been explored and plotted in Fig. 2g, Supplementary Fig. 12 and Supplementary Table 3[44]. In general, when the liquid metal inclusions without shear resistance replace the solid elastic polymer, the addition of liquid metal droplets will soften the composite. However, when the strong interfacial effects occur between the solids and liquid, such as the presence of a stiff oxide layer that forms at the liquid metal/Ecoflex interface (Supplementary Fig. 13), the composite will be effectively stiffened[44]. As shown in Fig. 2g, Supplementary Fig. 12 and Supplementary Table 3,

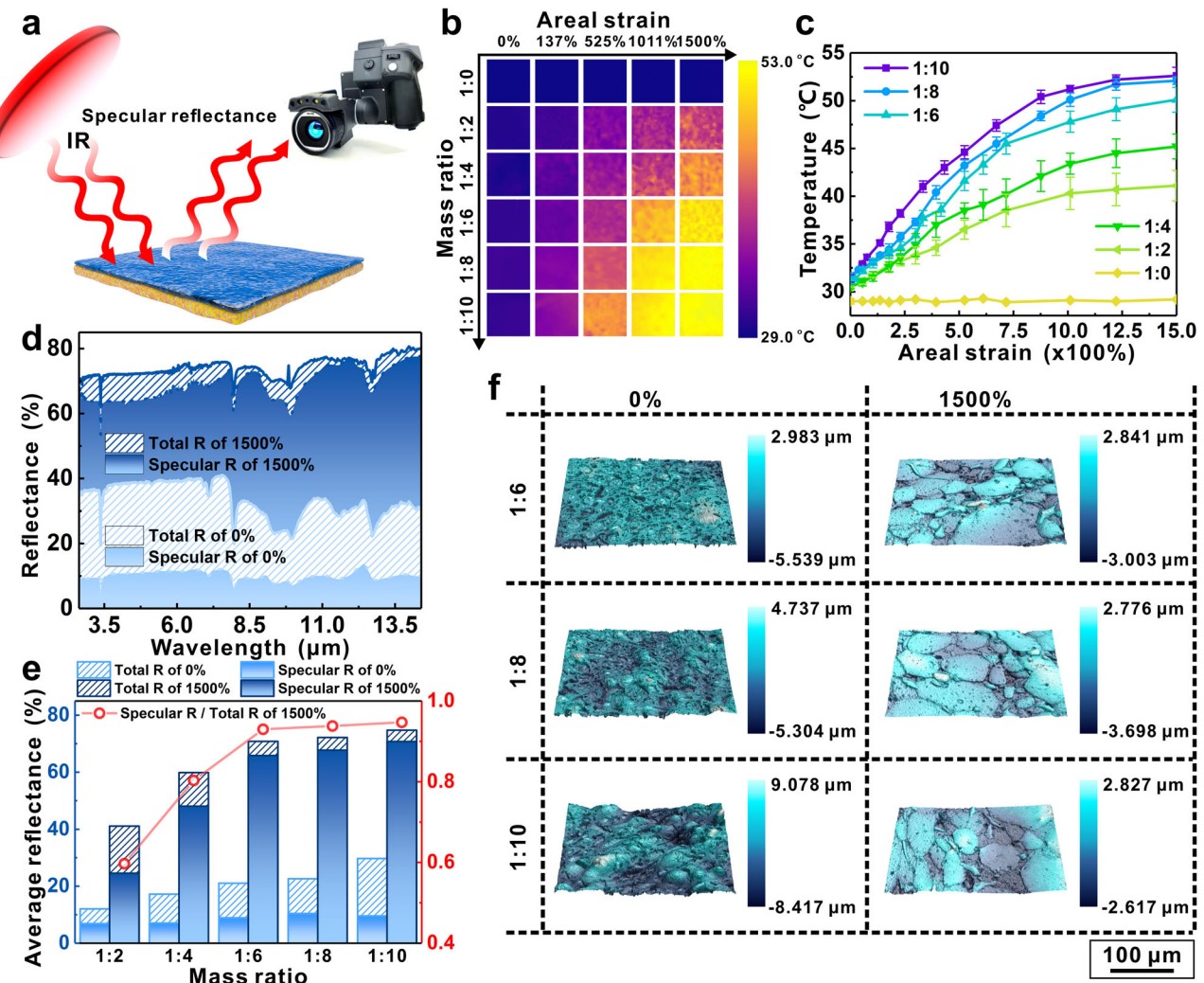

**Fig. 3 | Specular reflectance-based IR camouflage performance, specular reflectance spectra, and surface topography of BLEE film. a** Schematic illustration of the experimental setup of specular reflectance-based IR camouflage, where IR source (hot plate) is put on the left side and IR camera is put on the right side of the sample. **b** IR images of samples with different mass ratios of Ecoflex/EGaIn (1:0, 1:2, 1:4, 1:6, 1:8, and 1:10) under the applied areal strain of 0%, 137%, 525%, 1011%, and 1500%, respectively. **c** The temperature curves taken from the IR images of BLEE films in Fig. 3b. All error bars represent the standard deviation. **d** The total reflectance and specular reflectance spectra of BLEE films with mass ratio of 1:10, after applying different areal strains (0% and 1500%). **e** The calculation results of average total reflectances and specular reflectances in the range from 7.5 μm to 14 μm (histogram), and the ratios of average specular reflectance to total reflectance (red line). **f** The 3D surface topographies of BLEE films with mass ratios of 1:6, 1:8, and 1:10, after applying areal strains of 0% and 1500%. The scale bar reflects the intensity of surface fluctuation.

the Young's moduli of composites with lower liquid metal concentrations (Ecoflex/EGaIn mass ratio of 1:2, 1:4, 1:6) in Ecoflex are smaller than the pure Ecoflex. However, with the increase of liquid metal concentrations, Young's modulus gradually increases because the presence of the oxide layer inhibits the deformation of liquid metal droplets and stiffens the composites[44]. When the mass ratios increase to 1:8 and 1:10, their Young's moduli are even higher than that of pure Ecoflex. The reliability of IR camouflage performance has been also explored. The areal strain cycle test has been conducted on BLEE film with the mass ratio of 1:10. The average reflectance and IR camouflage performance (apparent temperature) have been measured to manifest excellent reusability and stability of BLEE film even after 2000 cycles (Fig. 2h and Supplementary Fig. 14). The results prove the samples have good stability without performance degradation. Moreover, such high stretchability of BLEE film enables it to wrap, and hide the IR signatures from the human body or other warm objects (Fig. 2i and Supplementary Fig. 15). Our proposed method of liquid metal-based IR-modulating materials is not only applicable in Ecoflex but also can be expanded to other polymer matrix, such as styrene ethylene

butylene styrene (SEBS), and polydimethylsiloxane (Supplementary Figs. 16, 17 and Supplementary Table 4). By using the SEBS polymer matrix with Young's modulus higher than Ecoflex, BLEE achieved higher total reflectance change (27.5%) under lower areal strain (137%) than the Ecoflex matrix, which proved BLEE could be potentially driven by electrical devices for practical modulation[7,18].

## Specular reflectance-based IR camouflage

The specular reflectance-based IR camouflage could be achieved by BLEE film with shape-morphing liquid metal microdroplets. To evaluate the IR camouflage performance in the case of specular reflectance, a hot plate was used as an IR source and placed on the left side of the samples (Fig. 3a). The IR images of BLEE films exhibit higher apparent temperature by increasing either the concentration of EGaIn or applied areal strain (Fig. 3b). Similarly, to gain an intuitive relationship between apparent temperature and mass ratio or areal strain, the temperature curves sampled from IR images of Fig. 3b are plotted in Fig. 3c. The maximum temperature difference in our work is recorded to be 21.2 °C, which is the highest value among the reported specular

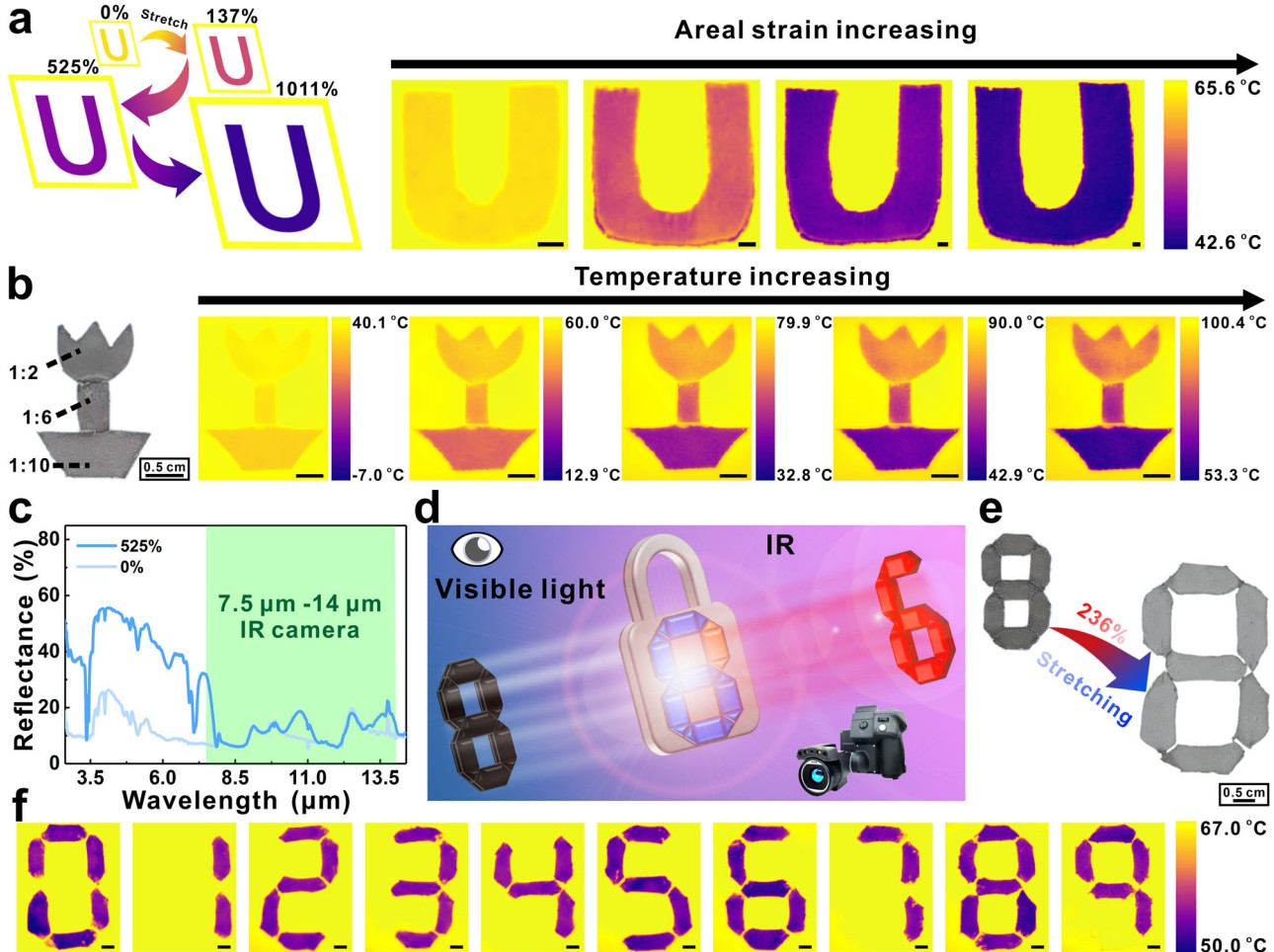

**Fig. 4 | Areal strain, and mass ratio control for IR ink, and encoding/decoding, respectively. a** Concept of changing the IR ink color when applying different areal strains (0%, 137%, 525%, and 1011%). The scale bar is 0.25 cm for IR images. **b** The optical image of the as-prepared "flower" pattern and IR images of the "flower" pattern placed on the hot plate under different temperatures. The scale bar is 1 cm for IR images. **c** Total reflectance spectra of the bottom layer of BLEE film with mass ratio of 1:10 before or after applying areal strain (green range represents the wavelength from 7.5 μm to 14 μm). **d** Concept of IR camera as a key to open the lock (decoding) and switch between visible light mode and IR mode. **e** The optical images of number pattern before and after applying areal strain. **f** IR images of the presentation of hidden numbers after applying a specific areal strain. The scale bar is 0.5 cm for IR images.

reflectance-based IR camouflage[6,7,18]. To gain mechanistic insights into the experimental results, the specular reflectance of BLEE films has been measured and analyzed. As shown in Fig. 3d and Supplementary Fig. 18, the specular reflectance of BLEE films is enhanced after applying the areal strain. To quantify the contribution of specular reflectance, the average specular reflectance and the ratio of specular to total reflectance have been calculated, respectively. Without stretching, the specular reflectance of BLEE films hardly changed, but the total reflectance slightly increases with the increased proportion of EGaIn (Fig. 3e). Noticeably, after applying 1500% areal strain, not only does the specular reflectance show an upward trend with the rising proportion of EGaIn, but the fraction of specular reflectance in the total reflectance grows as well. In this work, the maximum fraction of specular to total reflectance is 95% in the case of BLEE film with a mass ratio of 1:10. It results in nearly 61.2% of specular reflectance change, which is among the highest values reported[6,7,18].

Two factors can cause the change in specular reflectance. One is the deformation of the EGaIn droplets mentioned above, and the other is the surface roughness of the film. The 3D laser scanning microscope has been employed to identify the surface structure of BLEE film (Fig. 3f and Supplementary Fig. 19), and the corresponding representative height profiles along the red lines on optical images are shown in Supplementary Fig. 20. The surface roughness results and

analysis are summarized in Supplementary Table 5 (see more details in the Supplementary Materials). Compared with other published works that enhanced the specular reflectance by decreasing the surface roughness of polymer[7,18], the change of surface roughness of BLEE film measured in our work is small[7,18]. The deformation of EGaIn droplets instead of the surface roughness of film plays the most important role in tuning the specular reflectance.

### Multiple modes of IR regulation

Three parameters can affect the IR properties of these materials: (1) areal strain; (2) mass ratio, and (3) temperature. As for the areal strain, we have demonstrated that applying a specific areal strain is one of the important tools for tuning the IR total reflectance of BLEE film. In Supplementary Figs. 21 and 22 we used the precursor of Ecoflex and EGaIn mixture as IR ink to paint specific characters or symbols on polymer. Figure 4a and Supplementary Fig. 23 also show that The IR false color of character "U" changes under different applied areal strains.

As mentioned in the above section, the mass ratio of Ecoflex/ EGaIn is one of the main factors in adjusting IR total reflectance. By simply assembling parts with specific IR reflectance into a pattern, various false colors are present in the IR camera. The fabrication method is illustrated in Supplementary Fig. 24. A "flower" pattern was

designed and placed on a hot plate under different temperatures, the different parts of the "flower" were composed of the film with different Ecoflex/EGaIn mass ratios (Fig. 4b). To evaluate the false color of the pattern under different temperatures (Fig. 4b and Supplementary Fig. 25), the difference between the maximum and the minimum temperature of the scale bar in the IR images was set to be the same (see the normalization details in the Method). By applying the same areal strain, the color of the "flower" changed with the increasing hot plate temperature. Particularly, the chromatic distinction among the "flower" became larger with the rising of hot plate temperature. Such chromatic distinction is due to the different IR total reflectance of each part of the "flower" pattern (Supplementary Fig. 8). Further confirmed by the temperature difference calculation between the highest and lowest temperature in each IR image of Fig. 4b and Supplementary Fig. 25, the linear relationship (Supplementary Fig. 26) is consistent with the results in Supplementary Fig. 7b.

Apart from changing the mass ratio of BLEE film, the bottom layer (pure Ecoflex) of BLEE film can also exhibit distinct IR total reflectance compared with the top layer (Ecoflex/EGaIn mixture). As shown in Supplementary Fig. 27, the bottom layer of the BLEE film displays an apparent temperature close to the hot plate temperature. The total reflectance (Fig. 4c) and absorptance (Supplementary Fig. 28) of the bottom layer show that the polymer exhibits low reflectance and high absorptance invariably in the range from 7.5 μm to 14 μm with or without applying areal strain. Therefore, as shown in Fig. 4d, the double sides of BLEE films with different IR total reflectance can be assembled and used in visible/IR encoding/decoding systems. IR camera is the key to decode the switching between visible light and IR. In this work, the top layer (Ecoflex/EGaIn) and bottom layer (pure Ecoflex) assembled and displayed the number "8" under visible light mode (Fig. 4e and Supplementary Fig. 29). However, when the encoded numbers were placed on a hot plate and observed through an IR camera, the hidden numbers appeared due to different IR total reflectance of each part (Fig. 4f, Supplementary Fig. 30 and Supplementary Movie 5).

The IR behavior of liquid and solid metal particles differ. Thus, it is possible to tune the composition and thus, mp of metal alloys to program the temperature-dependent IR response. In this work, liquid metals with mp of 47 °C (Bi-Pb-In-Sn-Cd alloy) and 70 °C (Sn-Bi-Pb alloy) have been utilized to fabricate similar bilayered films to verify programmable IR painting and writing. As shown in Fig. 5a, b, the samples made from 70 °C (Sn-Bi-Pb alloy) exhibit low IR reflectance at room temperature (r.t.) without stretching (r-0%). Applying an areal strain of 1011% at r.t. (r-1011%) increases significantly the reflectance. However, when the sample was stretched above the melting temperature (m-1011%), the total reflectance became much higher than the sample stretched at r.t. (r-1011%). The IR images and apparent temperature were obtained by placing the samples on hot plates of 64 °C and 104 °C, respectively (Fig. 5c). The calculated apparent temperatures are well-matched to the measured ones (Supplementary Tables 6 and 7). The m-1011% sample performed the best IR reflectance (Fig. 5c). The optical and corresponding 3D laser scanning images of the samples' surfaces were taken to reveal the reason (Fig. 5d and Supplementary Fig. 31). Without applying areal strain (r-0%), alloy droplets were randomly dispersed in the polymer matrix. After applying areal strain (r-1011%), some alloy droplets transformed into a flake state, which was probably due to the supercooling of small alloy particles. However, other large alloy particles remained in the solid state, leading to higher surface roughness (Supplementary Table 8). When the samples were heated upon mp and then stretched (m-1011%), all the alloy droplets transformed into a flake state, leading to enhanced IR reflectance. The m-1011% sample was chosen to test mechanical stability. Figure 5e shows that there is no reduction in the average reflectance after 4 cycles, and the total reflectance spectra remain unchanged in Supplementary Fig. 32.

Considering the different mps of alloys, the temperature is used to program the IR pattern. A triangle pattern has been designed, in which each corner is made from specific alloy (16 °C (EGaIn), 47 °C (Bi-Pb-In-Sn-Cd alloy), and 70 °C (Sn-Bi-Pb alloy)) (Fig. 5f). With the rising of hot plate temperature, the part of alloy with lower mp than hot plate temperature experienced phase change, however, the part of alloy with higher mp remained in the solid state. Therefore, in the IR images, the color of the corners changed with the heating temperature (Fig. 5f). Such a system not only shows potential in programmable IR paintings but also shows broad application in anti-counterfeiting and human-machine interaction. Laser is another possible way to provide thermal energy[25]. As shown in Fig. 5g, an 808 nm laser was used as a "pen" to locally melt alloy. We used a laser to write "I, R" characters on the sample made from alloy with a mp of 70 °C. After stretching the sample and placing it on a hot plate, the patterns appeared.

## Multi-layered structural design

In prior works, IR encryption materials and systems often involve 2D designing, which merely allows 2D IR modulation and IR display by repeatedly adjusting the pixelated driver array[1,4,6,7,13]. In this work, we demonstrate a multi-layered structural design of IR-modulating systems to enhance the diversity of IR encryption systems and reduce their dependence on multiple drivers.

High mp metals including Au, Al, and copper (Cu) are usually evaporated on polymer film and widely used as IR-modulating materials[7,10,18,45]. Compared with the evaporated metal film-based IR-modulating materials, our BLEE film exhibits unique advantages such as liquid metal proportion depended IR reflectance, programmable IR display enabled by different low mp alloys, and multi-layered structural design by combining BLEE film with evaporated metal film together to achieve more complex IR patterns. Evaporated metal/polymer film shows high IR reflectance without applying an areal strain, but its reflectance reduces after being stretched. The stretching breaks the continuous metallic film and exposes the underlying polymer. In other words, the reflectance decreases with strain, whereas the films here increase reflectance with strain. These differences prompted us to develop a multi-layered programmable encryption structure by integrating high mp IR-reflecting metallic film and the EGaIn-based IR modulating film in one encryption system, in which various and more complex IR patterns can be generated and switched under visible or IR mode. In this work, evaporated Au film was chosen as a typical high mp metallic reflectance layer, and assembled with BLEE film to form a multi-layered structure (Fig. 6a). IR reflectance of pure Au film is shown in Fig. 6b, consistent with previous work. The programmable encryption method is demonstrated in Fig. 6c, high IR reflectance pattern made of Au film can be observed before stretching. In contrast, such Au film pattern disappears and the BLEE film pattern appears after stretching. Three kinds of patterns (contact, overlap, and partial overlap) were fabricated to qualify IR encryption performance (Fig. 6d and Supplementary Fig. 33). As expected, the IR pattern differed before and after stretching. For example, in the partial overlap design, the "fish" body was partially covered by the "seaweed" before stretching. After stretching, the "fish" body fully appeared in the IR image, indicating the ability to encrypt information (see more details in the Supplementary Materials).

To further enhance the diversity of IR encryption, Au film and BLEE films with different mass ratios of Ecoflex/EGaIn have been designed and stacked along the z direction (out-of-plane), as shown in Fig. 7a. BLEE film with different mass ratios (1:2 and 1:6) overlapped with Au film to form an H-shape pattern (Fig. 7a). With the increasing areal strains, the IR reflectance of Au film decreased, while the reflectance of BLEE-1:2 and BLEE-1:6 film continuously increased. It is worth noting that the clear patterns of BLEE-1:2 and BLEE-1:6 films appeared and were captured by IR camera under different areal strains of 525% and 178%, respectively. A programmable IR Morse coding can be designed in light of our multi-layered IR-modulating approach. For instance, five different areal strains (0%, 178%, 334%, 525%, and 751%)

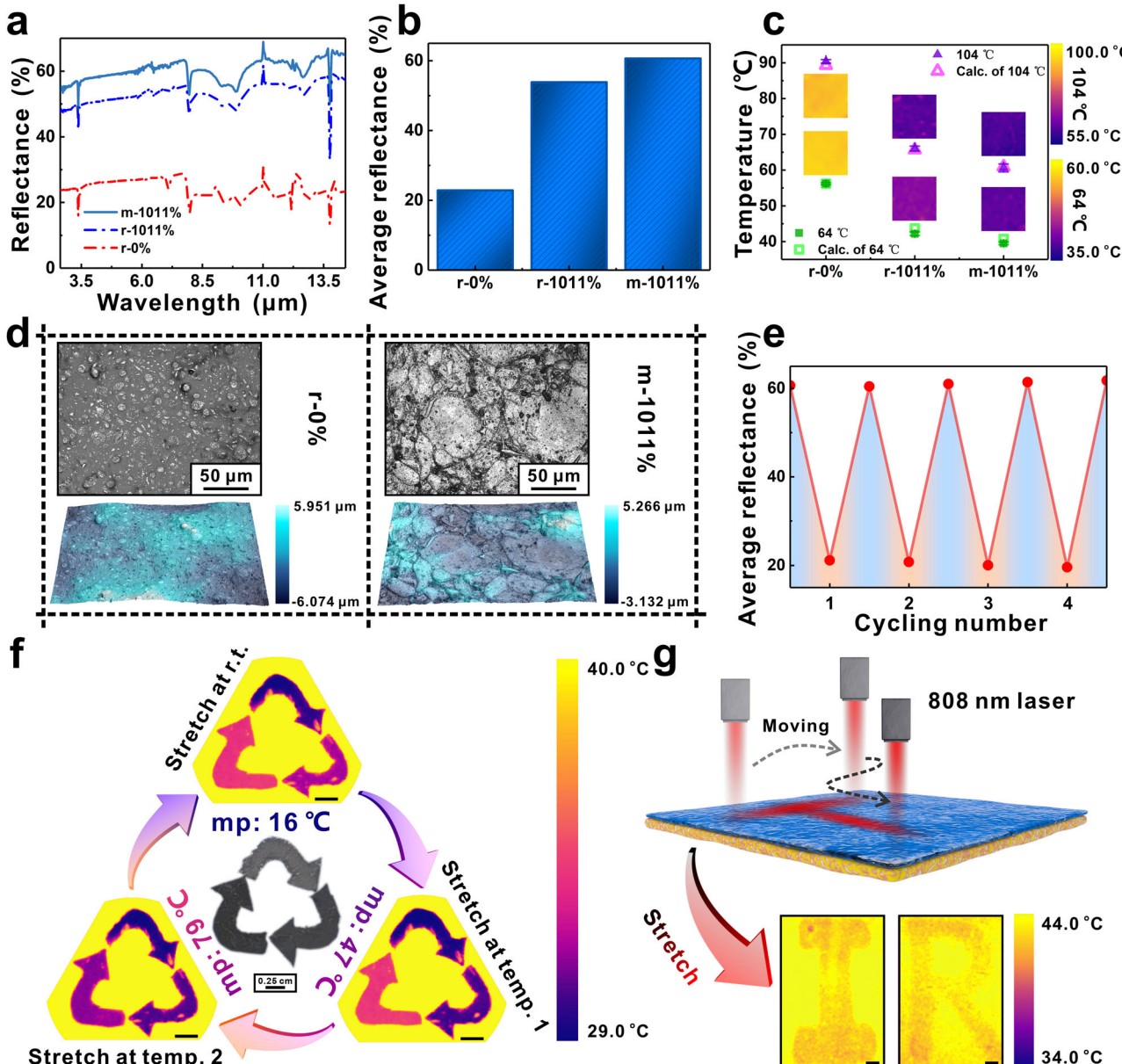

**Fig. 5 | Temperature control for IR painting, and writing. a** The total reflectance spectra of samples (r-0%, r-1011% and m-1011%) made from 70 °C (Sn-Bi-Pb alloy). **b** The average reflectances calculated from total reflectance spectra of samples (r-0%, r-1011% and m-1011%) made from 70 °C (Sn-Bi-Pb alloy). **c** The apparent temperatures (lines) and the calculated temperatures (dots) of samples (r-0%, r-1011% and m-1011%) made from 70 °C (Sn-Bi-Pb alloy) placed on the hot plate with a temperature of 64 °C and 104 °C, and the corresponding IR images are also displayed. **d** The optical images and 3D surface topography images of samples' surfaces (r-0% and m-1011%) made from 70 °C (Sn-Bi-Pb alloy). The scale bar reflects the intensity of surface fluctuation. **e** The average reflectance calculated from total reflectance spectra of the sample made from 70 °C (Sn-Bi-Pb alloy) with and without applying areal strain after different cycles (1, 2, 3, and 4). **f** The IR images of stretched triangle pattern under room temperature (r.t.), temperature 1 (temp. 1: 60 °C), and temperature 2 (temp. 2: 100 °C). The scale bar is 1 cm for IR images. **g** Schematic illustration of using a laser to write "I, R" character on Ecoflex/low mp alloy film made from 70 °C (Sn-Bi-Pb alloy) and corresponding IR images. The scale bar is 0.25 cm for IR images.

have been applied to four pre-designed multi-layered IR reflecting samples. It is worth noting that the key of the decryption process requires extracting information from IR images under pre-determined areal strains (0%, 334%, and 751%). The four programmable IR Morse codes are decrypted into letters L, O, V, and E, respectively (Fig. 7b).

## Discussion

In summary, we have developed a toolkit to dynamically change the IR reflectivity of a material by using stretchable liquid metal microdroplets embedded in a film of silicone elastomer. These films can modulate IR reflectivity using strain, composition, and temperature. Inspired by the dynamic nature of chameleon skin, liquid metal droplets have been utilized as deformable "melanophores" to adjust the intensity of IR reflectance and modify the false color observed through IR camera. Ecoflex matrix, similar to hormone, can drive the deformation of liquid metal droplets. Two different kinds of IR camouflage (total and specular reflectance) have been achieved by BLEE film. In the case of total reflectance-based IR camouflage, BLEE film can achieve 44.8% of total reflectance change and 34.5 °C of apparent temperature difference compared with the cloaked high-temperature object. Furthermore, by stretching, nearly 61.2% of specular reflectance change and nearly 21.2 °C of apparent

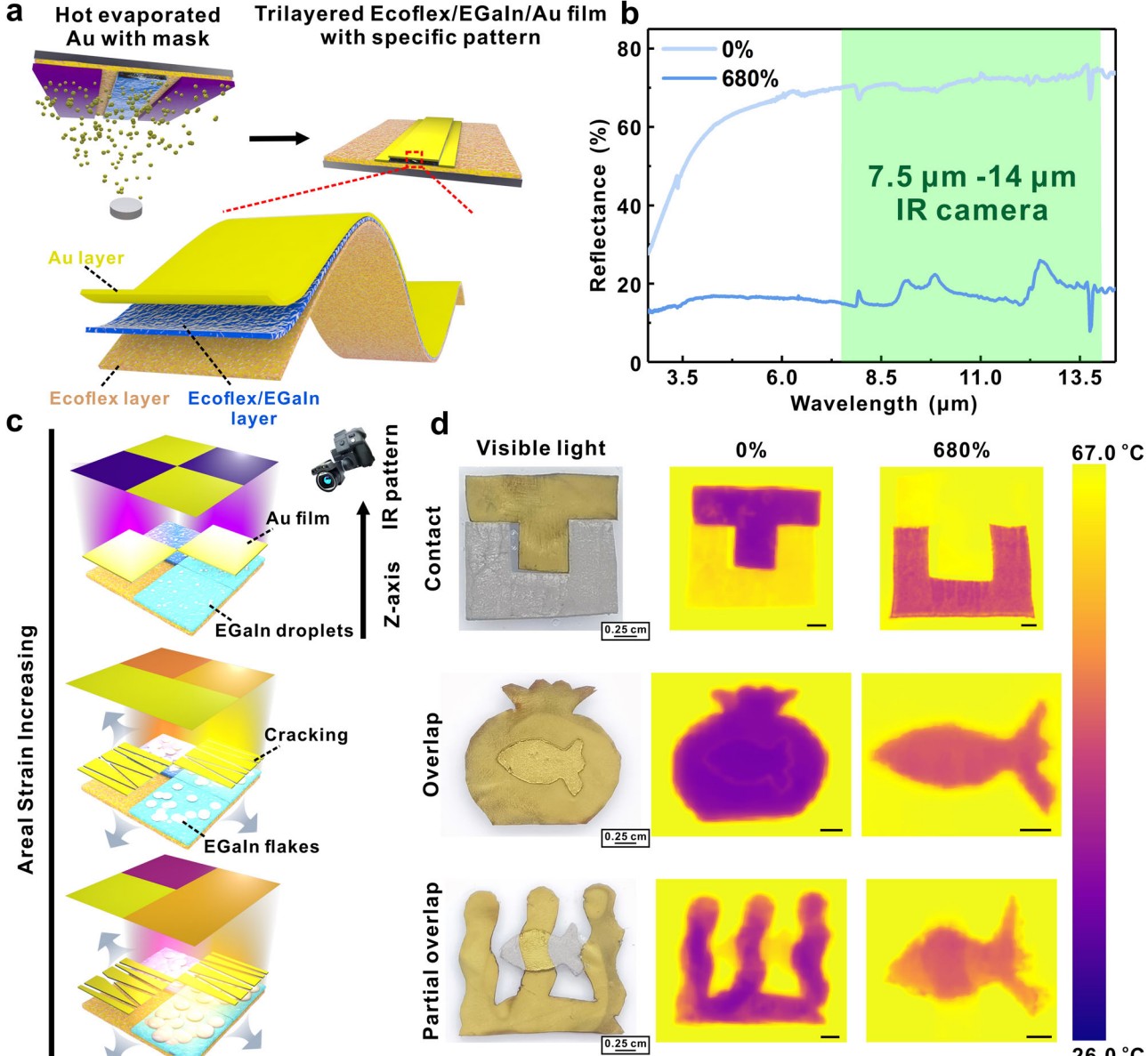

**Fig. 6 | Multi-layered structural design of EGaIn-based programmable IR encryption film. a** Schematic illustration of the fabrication process of trilayered Ecoflex/EGaIn/Au film. **b** The total reflectance spectra of Ecoflex/Au film with or without applied areal strain (green range represents the wavelength from 7.5 μm to 14 μm). **c** Concept of programmable IR encryption by stretching trilayered Ecoflex/ EGaIn/Au film. The red, yellow, green, and blue color represent the captured color in IR camera. **d** The optical and IR images (before and after stretching) of three kinds of patterns (contact, overlap, and partial overlap) fabricated by trilayered Ecoflex/EGaIn-1:10/Au film. The scale bar is 0.25 cm for IR images before stretching and the scale bar is 0.5 cm for IR images after stretching.

temperature change are obtained, which is among the highest values reported. Moreover, multiple modes of IR regulation including applying different areal strains, designing specific mass ratio of Ecoflex/EGaIn, and changing the surrounding temperature have been employed to encode/decode information between visible and IR modes and achieve IR painting/writing. Furthermore, a multi-layered design of IR modulating system has been developed to increase the diversity of IR encryption patterns, enhance the information density of stored patterns, and improve information security compared with regular 2D IR encryption systems. We expect that such rationally designed shape-morphing liquid metal droplets in elastomer can not only serve as potential IR camouflage, encoding/encoding, painting/writing, and encryption functional materials, but also expand its scope to other IR-related fields including sensing, thermal management, actuation, flexible electronics, artificial intelligence, and biomedicine.

## Methods

### Materials
Eutectic gallium indium alloy (EGaIn, 75.5 wt% gallium, and 24.5 wt% indium) was purchased from Shenyang Jiabei Commercial Trading Company. Low mp alloy with mp at 47 °C (Bi: 44.7 wt.%; Pb: 22.6 wt.%; In: 19.1 wt.%; Sn: 8.3 wt.%; and Cd: 5.3 wt.%) and 70 °C (Sn: 25–30 wt.%; Bi: 50–60 wt.%; and Pb: 20–25 wt.%) were purchased from Dongguan Dingtai Commercial Trading Company. Ecoflex 0030 (Smooth-On, Inc.) was purchased from Shanghai Fanhe Commercial Trading Company. Hexane (99%) was purchased from Adamas-beta.

### Fabrication of BLEE
Ecoflex precursor was prepared by mixing the two parts of Ecoflex 0030 with the mass ratio of 1:1. Such mixture was shaved on a flat acrylic plate and cured at room temperature. Ecoflex precursor and EGaIn were mechanically mixed at different mass ratios of (1:2, 1:4, 1:6,

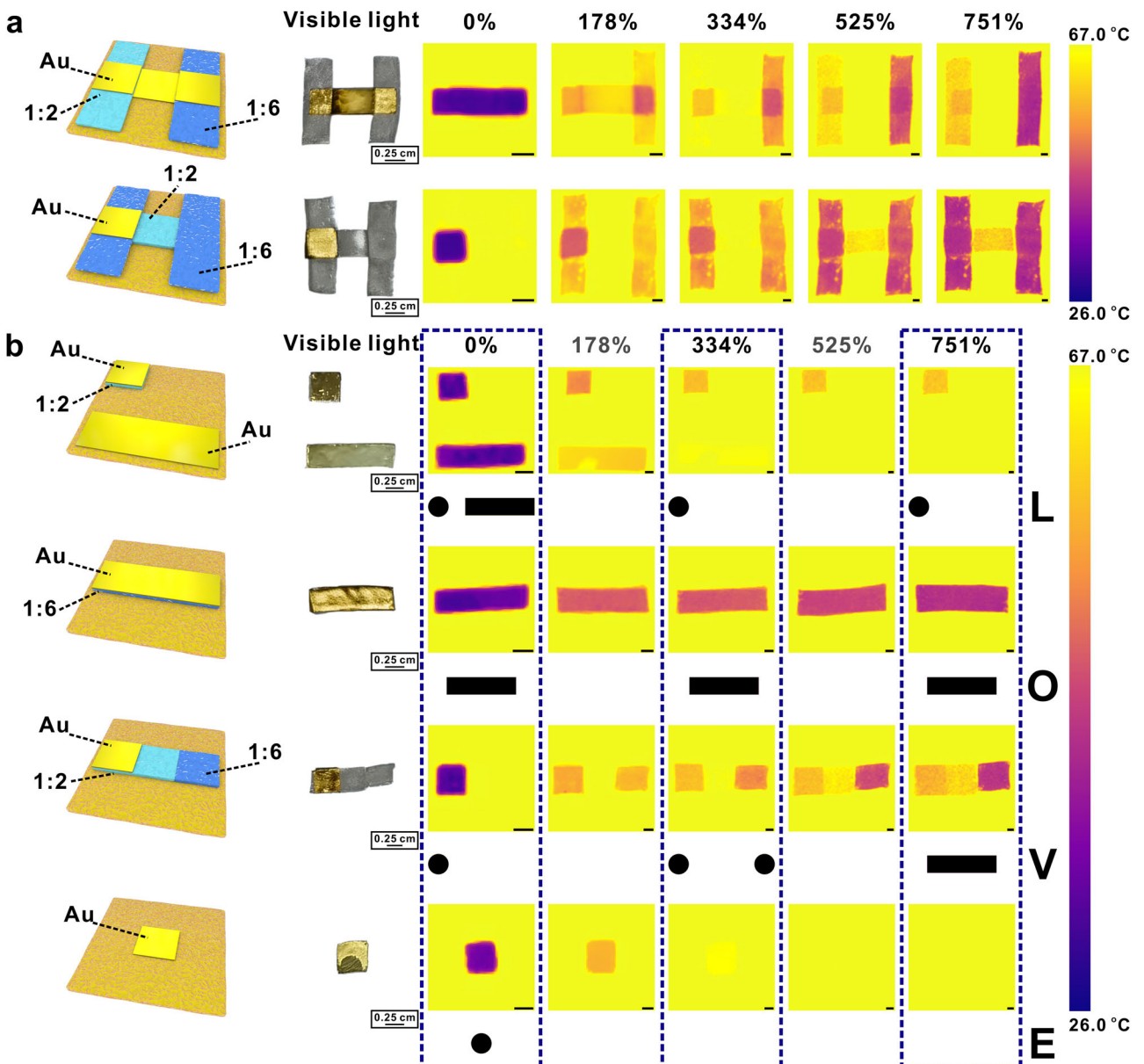

**Fig. 7 | Demo of programmable IR Morse coding. a** The optical and IR images (with applied areal strains of 0%, 178%, 334%, 525%, and 751%) of two types of patterns fabricated by trilayered Ecoflex/EGaIn-1:2&1:6/Au film. The scale bar is 0.25 cm for IR images. **b** Demo of programmable IR Morse coding by applying pre-determined areal strains on the specifically designed trilayered Ecoflex/ EGaIn-1:2&1:6/Au film patterns. The scale bar is 0.25 cm for IR images.

1:8, and 1:10) at 700 rpm for 5 min. Ecoflex/EGaIn mixture was shaved on the top of pure Ecoflex layer and cured at room temperature. The thickness of as-prepared BLEEs is shown in Supplementary Table 9.

**Note 1:** The single-layered Ecoflex/EGaIn film was fabricated by using the same process, but the mixture was directly shaved and cured on a flat acrylic plate at room temperature.

**Note 2:** To obtain the liquid metal droplets within Ecoflex, Ecoflex precursor was first mechanically mixed with EGaIn at different mass ratios (1:2, 1:4, 1:6, 1:8, and 1:10) at 700 rpm for 5 min, which is the same as the fabrication process of BLEE. After mixing, 5 ml hexane was added to the mixture and stirred at 300 rpm for 10 min. The Ecoflex precursor gradually dissolved in the hexane solvent. The mixture was centrifuged at 409 g for 10 min, and the sediment was redistributed in hexane. After repeating the above purification steps twice, the Ecoflex was completely removed. Finally, the liquid metal particles dispersed in the hexane were dropped on the cleaned silicon wafer to prepare the samples for SEM analysis.

**Fabrication of bilayered Ecoflex/low mp alloy film**
Ecoflex 0030 precursor was first shaved on an acrylic plate and cured with the same method as before. Low mp alloy was heated up and mechanically mixed with one of the Ecoflex precursor at 700 rpm for 5 min so that the low mp alloy was kept in a liquid state and the Ecolfex would not cure during mixing. After the mixture was cooling down, it would be mixed with another Ecoflex precursor. The mixture was shaved on the top of pure Ecoflex layer and cured at room temperature. In Supplementary Fig. 34, the differential scanning calorimeter (DSC) curves showed the mps of Ecoflex/low mp alloy (mp at 47 °C, and 70 °C) films were 48.0 °C and 70.9 °C, respectively.

**Fabrication of trilayered Ecoflex/EGaIn/Au film with different patterns**
A mask cut from cardboard was used to make contact, overlap or partially overlap on the BLEE pattern. Au film of ~200 nm thick was deposited on the mask/pattern by vacuum thermal evaporation

apparatus (JSD400, Jiashuo Vacuum Technology Co., Ltd, China). After taking the mask off, trilayered Ecoflex/EGaIn/Au film was prepared.

## Instrumentation

The structure and morphology of the as-prepared Ecoflex/EGaIn films were characterized by scanning electron microscope (SEM, Quanta, FEI, USA) with an accelerating voltage of 10 kV and optical microscope (UCMOS 14000KPA, TOUPTEK PHOTONICS, China) with Toup View software package. The thickness of the gallium oxide layer was characterized by transmission electron microscope (TEM, Talos F200X G2, Thermo Scientific, USA) with an accelerating voltage of 200 kV. The optical images were taken by a visible camera (Mate 40 Pro, Huawei, China). The 3D reconstructor analysis was characterized by X-ray microscope (XRM, Xradia 520 Versa, Carl Zeiss, Germany). The roughness and 3D structure of the surface of films were characterized by a 3D Laser Scanning Microscope (LSM, VK-X3000 series, Keyence, Taiwan, China). The IR images and videos were taken by IR camera (T640, FLIR, USA) and the results were analyzed by Research IR and iMovie software packages. The mechanical properties of samples were characterized by a dynamic mechanical analyzer (DMA, 850, TA, USA). The thickness of samples was characterized by a film thickness gauge (KEJIA CO., LTD., China). The mp of the Ecoflex/low mp alloy mixture was tested by Differential Scanning Calorimeter (DSC, 2500, TA, USA). The IR properties of samples were characterized by Fourier transform infrared spectrometer (FTIR, Nicolet 6700, Thermo Fisher Scientific, USA) outfitted with Pike Technologies Mid-infrared Integrating Sphere. The specular reflectance of samples was characterized by an infrared imaging microscope (Nicolet iN10 MX, Thermo Fisher Scientific, USA).

## IR camera visualization of Ecoflex/EGaIn films

To characterize the influence of total reflectance on the IR camouflage, BLEE films with different Ecoflex/EGaIn mass ratios of 1:0, 1:2, 1:4, 1:6, 1:8, and 1:10 were separately put on a hot plate with a temperature of 76 °C under specific areal strains. The IR camera was placed directly above the hot plate, as shown in Fig. 2a. The IR images of different BLEE films with specific areal strains (0%, 137%, 525%, 1011%, and 1500%) were selected and shown in Fig. 2b, where the images were cut into the size of 10 mm × 10 mm. The highest temperature of the scale bar was set to be 73 °C, which was the highest temperature among all the images. The lowest temperature of the scale bar was set to be 41 °C, which was the lowest temperature among all the images. The average temperature and error bar were taken from the IR images of each sample, and the data was obtained through the Research IR software package. In Fig. 2c, the slope of apparent temperature curves of BLEE films is steep at the initial state, and they gradually reach a plateau with the increase of areal strain. When the EGaIn mass ratio increases to 1:8, the lowest apparent temperature hardly changes, and reaches a threshold. To evaluate the influence of hot plate temperature on the IR camouflage, BLEE film with different Ecoflex/EGaIn mass ratios under specific areal strain (1500%), pure Ecoflex, and EGaIn were placed on the plate separately. By switching the temperature of the hot plate (41 °C, 46 °C, 51 °C, 56 °C, 61 °C, 66 °C, 71 °C, 76 °C, 81 °C, 86 °C, 91 °C, 96 °C, 101 °C, and 106 °C), the corresponding IR images were obtained. As for the films made from low mp alloy (mp: 70 °C), the IR images and average temperature of films under the areal strain of 1500% in Fig. 5c were collected by the same method. However, the temperature of the hot plate was set to be 64 °C and 104 °C, separately.

To characterize the influence of specular reflectance on the IR camouflage, BLEE films with different Ecoflex/EGaIn mass ratios of 1:0, 1:2, 1:4, 1:6, 1:8, and 1:10 were separately put on a desk without heating under specific areal strains. The hot plate with a temperature of 76 °C was placed on the right side of the samples, and the IR camera was placed on the left side of the samples. The angle between the hot plate and the IR camera should satisfy the condition of specular reflectance, as shown in Fig. 3a. The IR images and the average temperature were collected by the same method as above. The highest temperature of the scale bar was set to be 53 °C, which was the highest temperature among all the images of samples. The lowest temperature of the scale bar was set to be 29 °C, which was the lowest temperature among all the images of samples. In Fig. 3c, BLEE films with higher volume of EGaIn exhibit greater slope of apparent temperature curves in the function of areal strain, but temperature curves of all the BLEE films become flat when the areal strain exceeds 1000%.

## Stability testing of Ecoflex/liquid metal films

The mechanical stability of films was demonstrated in the following way. Firstly, the films with a mass ratio of 1:10 (Ecoflex/EGaIn) were cycled under areal strain between 0% and 300% for different times (5, 100, 200, 400, 1000, and 2000). Secondly, the samples after cycling and the sample without applying areal strain were both stretched to 1500%. The samples were separately placed on the hot plate (76 °C) and were taken IR images. The total reflectance of samples was obtained by FTIR, and the $R_{average}$ was calculated using the above method.

The mechanical stability of films made from low mp alloy (mp: 70 °C) was also performed. The film was put on the hot plate to melt the alloy droplets in the Ecoflex. Such heated film was stretched under the areal strain of 1011%. The stretched film was re-heated and the strain was removed. The whole test process was repeated four times. The total reflectance of the sample with applied areal strain and without applied areal strain was measured. And the $R_{average}$ was also calculated by using the same method.

## Areal strain control for "SJTU" pattern

Ecoflex precursor was shaved on a flat acrylic plate and cured at room temperature. A mask with specific pattern was designed and put on the cured Ecoflex layer. The mixture of Ecoflex precursor and EGaIn with a mass ratio of 1:10 was shaved on the mask. After taking the mask off, the prepared pattern was cured at room temperature. The samples were applied with different areal strains (0%, 137%, 525%, and 1011%) and put on the hot plate with a temperature of 70 °C. The minimum value of the scale bar was the lowest apparent temperature of the sample under the largest areal strain (T1011%). The maximum value of the scale bar was the highest apparent temperature of the sample without applied areal strain (T0%) plus 0.2 times the difference between T1011% and T0%. The specific values of the scale bar are shown in Supplementary Table 10.

## Mass ratio control for "flower" pattern

BLEE films with different mass ratios of 1:2, 1:6, and 1:10 were cut into parts of a "flower" shape with a graver. Ecoflex precursor was first shaved on a flat acrylic plate. The as-prepared different parts of a "flower" shape were put on the uncured Ecoflex precursor and were cured together at room temperature. The petal, stem, and pot of the "flower" were made from BLEE films with Ecoflex/EGaIn mass ratios of 1:2, 1:6, and 1:10, respectively. The sample was stretched under an areal strain of ~809% and placed on the hot plate. After changing the temperature of the hot plate, the maximum temperature of the samples was recorded by an IR camera. When the maximum temperature was 100 °C, the largest difference between the highest temperature and the lowest temperature (47.1 °C) could be obtained through IR images. To normalize the changing of colors between three "flower" parts under the different temperatures of the hot plate, the difference between the highest temperature and the lowest temperature in the scale bar for all the IR images was set to be 47.1 °C. The specific values of the scale bar were shown in Supplementary Table 11.

## Mass ratio control for "numbers" pattern

BLEE film with a mass ratio of 1:10 was cut into a hexagonal shape with a graver. Ecoflex precursor was first shaved on a flat acrylic plate. Before the Ecoflex precursor was cured, the as-prepared BLEE hexagonal shapes were put on the uncured Ecoflex precursor to splice into an "8" character. In particular, if the top layer of the BLEE hexagonal shape is faced up, it is visible in the IR camera. If the bottom layer of BLEE film (pure Ecoflex) is faced up, it is invisible in the IR camera. The optical images of different "numbers" were taken by the optical camera. Samples with an applied areal strain of ~236% and samples without applied areal strain (0%) were put on the hot plate, respectively. Keep the temperature of the hot plate maintain 70 °C, and record the IR movies when putting the samples (areal strain: ~236%) on the hot plate. The scale bar of all the IR images and movies was set to be 50 °C–67 °C, where the highest and the lowest temperature of IR images were included.

## Temperature control for Ecoflex/low mp alloy

The triangle pattern Ecoflex/low mp alloy film was prepared by the method as same as the "flower" pattern. However, the three corners of the triangle were made from EGaIn (mp: 16 °C) and other two kinds of low mp alloys (mp: 47 °C or 70 °C), respectively. The triangle sample was stretched at room temperature, or stretched after heating by a hot plate with a temperature of 60 °C, or stretched after heating by a hot plate with a temperature of 100 °C. The areal strain was set to be ~756%. Then, the sample was put on the hot plate with a temperature of 42 °C as a background temperature for taking the IR pictures. The scale bar of all IR images was set to 29 °C–40 °C.

## IR laser writing via Ecoflex/low mp alloy

The Ecoflex/low mp alloy film for IR laser writing was made from low mp alloy (mp: 70 °C). The sample was placed on the table at room temperature. Then, using the 808-nm laser (VA-I-DC-808, Bangshou Technology Co., Ltd, Beijing) to locally heat the samples at a power density of ~1.4 W/cm². After heating, the sample was stretched under an areal strain of ~756% and placed on a hot plate with a temperature of 64 °C. The IR images were taken and the scale bar of all IR images was set to 34.0 °C–44.0 °C.

## Multi-layered structural design for trilayered Ecoflex/EGaIn/Au film

To fabricate the trilayered Ecoflex/EGaIn-1:10/Au film (contact, overlap, or partially overlap), the mixture of Ecoflex/EGaIn-1:10 was cured on the pure Ecoflex film with a mask first. Then, ~200 nm Au film was evaporated on the Ecoflex/EGaIn-1:10 pattern with a mask on it. As for the IR encryption pattern (contact, overlap, and partial overlap), trilayered Ecoflex/EGaIn-1:10/Au film with an applied areal strain of ~680% and samples without applied areal strain were put on the hot plate (70 °C), respectively. The scale bar of all the IR images was set to be 26 °C (room temperature) –67 °C.

To fabricate the trilayered Ecoflex/EGaIn-1:2&1:6/Au film and samples for Morse code, Ecoflex/EGaIn-1:2 and Ecoflex/EGaIn-1:6 films were cut into specific patterns and assembled as the design shown in Fig. 7a, b. Then, ~200 nm Au film was evaporated on the Ecoflex/EGaIn-1:2&1:6 pattern with a mask on it. For the patterns made from Ecoflex/Au, the patterns disappear in IR images when the areal strain is larger than 334%. For the patterns made from Ecoflex/EGaIn-1:2, the patterns display when the areal strain is larger than 525%. For the patterns made from Ecoflex/EGaIn-1:6, the patterns display when the areal strain is larger than 178%. For the patterns made from Ecoflex/EGaIn-1:2/Au and Ecoflex/EGaIn-1:6/Au, the patterns always display. As for the programmable IR encryption system for Morse code, trilayered Ecoflex/EGaIn-1:2&1:6/Au film with applied different areal strain (0%, 178%, 334%, 525%, and 751%) were put on the hot plate (77 °C), respectively. The scale bar of all the IR images was set to be 26 °C (room temperature) –67 °C.

## Reporting summary

Further information on research design is available in the Nature Portfolio Reporting Summary linked to this article.

## Data availability

All data generated in this study are presented in the paper and the Supplementary information. Source data are provided with this paper.

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

## Acknowledgements

We thank the Center of Hydrogen Science of Shanghai Jiao Tong University and the Zhiyuan Innovative Research Center (ZIRC) for providing scanning electron microscope, optical microscope, and IR camera. We thank the Instrumental Analysis Center of Shanghai Jiao Tong University for providing 3D laser scanning microscope, Fourier transform infrared spectrometer, dynamic mechanical analyzer, film thickness gauge, differential scanning calorimeter, and infrared imaging microscope. This work was supported by the following fundings: National Key R & D Project from Ministry of Science and Technology of China Grant (2022YFA1203100 (T.D.)), National Natural Science Foundation of China (51973109 (C.S.)), The Innovation Program of Shanghai Municipal Education Commission (2019-01-07-00-02-E00069 (T.D.)), The National Science Foundation (2032409 (M.D.D.)).

## Author contributions

C.S., T.D., M.D.D. and Y.Z., designed research. Y.Z., H.Z., S.A. and W.X. performed fabrication, testing, and characterizations. C.S., T.D., M.D.D., Y.Z., H.Z., S.A., W.X., B.F., P.T., W.S. and J.W. discussed and analyzed the results and contributed to the writing of the paper.

## Competing interests

The authors declare no competing interest.
