## [Peer Review File · Nature Communications]

Chameleon-Inspired Tunable Multi-layered Infrared-Modulating System via Stretchable Liquid Metal Microdroplets in Elastomer FilmREVIEWER COMMENTS

Reviewer #1 (Remarks to the Author):

“Chameleon-Inspired Tunable Three-Dimensional Infrared-Modulating System via Stretchable Liquid Metal Microdroplets in Elastomer Film” submitted by Zhang et al. reports their study on liquid metal-based infrared-modulating materials for dynamic IR systems. It is shown that the mid-infrared radiation can be modulated by changing areal strain, mass ratio of the constituent Ecoflex/EGaIn and temperature of the device under concern, whereas incorporation of different liquid metals with distinct melting temperatures and trilayer configuration using additional metal layer enables more diverse applications. A few interesting demonstrations such as IR ink encoding/decoding are presented, however, I find several issues that should be addressed before considering this study for publication.

1. It is stated that BLEE is inspired by the mechanism of chameleon skin, but it is not fully convincing at the moment: As it is depicted in Figure 1, brightness change in some specific chameleon may result from movements of melanosomes in response to hormonal stimulation, [Zeitschrift für Zellforsch. und Mikroskopische Anat. 104, 282–294 (1970)] whereas the modulation in BLEE is obtained by applying areal strain. More notable characteristics of chameleon is change in reflective color, and it is indeed obtained by period modulation of the lattice [Nat. Commun. 6, 1–7 (2015)] which is possibly related to the areal strain. In this regard, it appears that the type of stimulation and modulation mechanism of the current BLEE are not perfectly matched to the chameleon skin, and I suggest reconsideration of the counterpart found in nature.

2. Dynamic IR regulation of BLEE is studied in the range of 0~1,500 % of strain, and this number is considerably large. Although strain-dependent IR regulation is an interesting concept that stems from biological systems, there is no need to follow exact anatomies of the color-changing animals. In this regard, areal strain, for such high value, might not be practical for dynamic systems. In other similar studies, the required strain was considerably smaller, e.g. Science. 359, 1495-1500 (2018), ACS Nano. 15, 17299-17309 (2021), and it was capable to prepare electrically actuated camouflage devices for practical modulation. Recent study on IR regulator also demonstrated electrically controlled device.[Nat. Commun. 14, 5087 (2023)]

3. Here are some other comments that you may consider:

- The internal structure of BLEE remains unclear: Figure 1B illustrates that EGaIn exists in the form of discrete droplets inside Ecoflex matrix and so they are after applying areal strain, but it is presented that Ecoflex and EGaIn are mechanically mixed, according to Materials and methods section. Since the ratio of Ecoflex/EGaIn can be as high as 1:10, it is anticipated that EGaIn will create quasi-continuous layer within Ecoflex matrix, which will lead to the discussion that illustration in Figure 1B can be misleading. Although many SEM and optical images of BLEE are presented in Figures and Supplementary Information, more detailed structure from topological perspective should be clarified. Cross-section images also should be more informative. These investigations will help the authors to conduct more in-depth theoretical analysis on the current system, which is relatively weak in the current version.

- Mechanical properties of BLEE films at different mass ratios are shown in Figure 2G, and it shows that the modulus of BLEE at high EGaIn portion is higher. Why?

- In Figure 5, it is mentioned that EGaIn-based programmable IR encryption film is realized in 3D structure, but this expression is not very convincing: by adding third layer, now the device is in trilayer configuration, but increase in thickness in out-of-plane direction is almost insignificant. It is shown that overlapped information can be distinguished by adding the third layer, yet I would like to recommend the authors to find some different term.

- Please check again for minor mistakes, e.g. “high-reflectacne state” in Figure 1E, etc.

Reviewer #2 (Remarks to the Author):

In this work, Zhang et al. presented liquid metal-based infrared-modulating materials and systems with multiple modes to regulate the infrared reflection. This work seems interesting but the innovation needs to be improved. Therefore, I recommended it to be published after some revisions with the following comments:

1. This material must be stretched to change the infrared reflectance, so what is its specific application scenario?
2. A similar principle of changing the reflectance has been reported. What are the advantages of your work? (Materials Today Chemistry 24 (2022) 100911)

Reviewer #3 (Remarks to the Author):

The manuscript presents a layered structure consisting of liquid metal droplets in silicone elastomer (Ecoflex), that allows IR modulation through IR reflectance. The deformation of liquid metal microdroplets enables the IR reflectance changing from a relatively low state to a high reflectance state, which is regulated under mechanical strain. This is a proof-of-concept study showing the potential utility of EGaIn/Ecoflex composites as IR camouflage; however, the presented results are not sufficient to support the major claims of the study. Thus, with major revisions this work would be better suited for a more specialized journal.

1. The primary concern in the present manuscript revolves around the insufficient characterization of the liquid metal droplet structure within the composite. While SEM analysis was conducted, it lacks detailed information about the liquid metal (LM) droplets, such as their size when embedded in the composite.
2. Moreover, it is essential to investigate whether there are any size variations in LM droplets at different EGaIn/Ecoflex ratios.
3. The major claim of the paper hinges on the stretching of the composite, resulting in the deformation or shape change of the EGaIn droplets within the elastomer. Unfortunately, this shape change has not been adequately characterized in the manuscript, thereby leaving a gap in the support for the paper's major claim.
4. Additionally, there is a lack of clarity regarding the relationship between the size of EGaIn droplets and the regulation of IR-reflectance. It would be beneficial to elucidate the specific size or size range at which IR-reflectance becomes more significant in the context of the study.
5. Is there any surface oxidation observed in the liquid metal droplet? The manuscript lacks information on the surface properties of the LM droplets and how these properties might influence the IR reflectance of the composites. Providing insights into the surface characteristics of the liquid metal droplets and their potential impact on IR reflectance would contribute to a more comprehensive understanding of the study.
6. Once more, the authors have presented characterizations of the film's surface roughness using optical imaging. However, it remains uncertain whether the tuning of specular reflectance was solely attributable to the deformation of the LM droplets, as there is a lack of supporting data. It is possible that the observed effect is linked to the formation and deformation of the LM droplets network within the elastomer matrix (Science378,637-641(2022).DOI:10.1126/science.abo6631). Providing additional data or clarification on this aspect would strengthen the argument and enhance the overall credibility of the findings.
7. Furthermore, the rationale behind choosing Ecoflex over alternative elastomers, such as PDMS, remains unclear. It would enhance the comprehensibility of the study to include a comparison or discussion addressing the specific reasons for opting for Ecoflex and how it compares to other elastomeric materials.

RESPONSE TO REVIEWERS' COMMENTS

Dear Reviewers,

We would like to thank all of you for your time and effort in reviewing our manuscript. We appreciate all your comments that help further improve the quality of our work. Following is our point-to-point responses to your comments:

Comments from Reviewer #1

“Chameleon-Inspired Tunable Three-Dimensional Infrared-Modulating System via Stretchable Liquid Metal Microdroplets in Elastomer Film; submitted by Zhang et al. reports their study on liquid metal-based infrared-modulating materials for dynamic IR systems. It is shown that the mid-infrared radiation can be modulated by changing areal strain, mass ratio of the constituent Ecoflex/EGaIn and temperature of the device under concern, whereas incorporation of different liquid metals with distinct melting temperatures and trilayer configuration using additional metal layer enables more diverse applications. A few interesting demonstrations such as IR ink encoding/decoding are presented, however, I find several issues that should be addressed before considering this study for publication.”

Response: Thanks to the reviewer for the encouraging comments that help to strengthen the quality of our manuscript. We have revised our manuscript and carried out additional experiments to address the concerns of the reviewer.

Comment 1:

“It is stated that BLEE is inspired by the mechanism of chameleon skin, but it is not fully convincing at the moment: As it is depicted in Figure 1, brightness change in some specific chameleon may result from movements of melanosomes in response to hormonal stimulation, [Z. Zellforsch. 104, 282–294 (1970)] whereas the modulation in BLEE is obtained by applying areal strain. More notable characteristics of chameleon is change in reflective color, and it is indeed obtained by period modulation of the lattice [Nat Commun. 6, 6368 (2015).] which is possibly related to the areal strain. In this regard, it appears that the type of stimulation and modulation mechanism of the current BLEE are not perfectly matched to the chameleon skin, and I suggest reconsideration of the counterpart found in nature.”

Response: Thanks to the reviewer for the helpful comment. We apologize that we didn't clarify the similarity between the brightness change mechanism of chameleon skin and our bilayered Ecoflex/EGaIn (BLEE) infrared (IR) modulation mechanism. As mentioned by the reviewer, our BLEE IR modulating materials employed areal strain rather than hormonal stimulation to achieve brightness change. However, both of these two cases exhibit similar physical principle of modulating light reflection. There are following reasons for choosing chameleon skin as the counterpart in nature in this research:

Our BLEE is inspired by the dispersion and aggregation of melanosomes in chameleon skin. This physiological color change not only exists in panther chameleons, but also in other vertebrates, such as *Anolis carolinensis* (a kind of lizard) [*Zeitschrift für Zellforschung und Mikroskopische Anatomie* 104.2, 282-294 (1970)], frogs [*The Journal of Cell Biology* 38.1, 67-79 (1968)], and fishes [*American Zoologist* 9.2, 465-478 (1969)]. Many vertebrates change skin brightness by modulating the relative position of melanophores and iridophores. Under the pale background, melanosomes aggregate in a perinuclear position within melanophores, therefore, sunlight can directly interact with iridophores. Under the dark background, melanosomes migrate into the terminal processes of the melanophores, and overlay the iridophores. Thus, melanosomes effectively prevent sunlight penetration and hitting the iridophores below. We emphasize that this movement of melanosomes is similar to our shape-morphing liquid metal droplets. At the low-reflectance state, spherical liquid metal droplets (similar to the aggregated melanosomes) are encapsulated by the polymer. IR penetrates through the gap between liquid metal droplets and is absorbed by the bottom polymer layer. At the high-reflectance state, the polymer becomes thinner under stretching, and liquid metal droplets turn into flakes (similar to the dispersed melanosomes) that strongly reflect IR radiation and prevent IR absorption by the bottom polymer layer.

1

Shanghai Jiao Tong University, 800 Dong Chuang Road, Shanghai, 200240 P. R. China
Phone +86 (021) 5474-5582 / Fax +86 (021) 3420-2749 / dengtao@sjtu.edu.cn

We notice that the tunable brightness of chameleon skin exhibits a similar principle to the fake color in an IR camera. The previous work shows that chameleon shifts color through active tuning of a lattice of guanine nanocrystals within a superficial thick layer of dermal iridophores [*Nat Commun.* 6, 6368 (2015)]. Such a color adjustment process involves switching the wavelength of reflected light. Different from the above color change mechanism, the brightness change of chameleon skin would not change the wavelength of reflected light but change the intensity of reflected light. It is similar to the working principle of an IR camera that uses fake color to represent the light intensity of IR ranging from 7.5 to 14 μm . The fake color in the IR camera represents the integrated intensity of IR radiation from 7.5 ~ 14 μm . By applying various areal strains, the BLEE can reflect the incident IR with different intensities and exhibit tunable fake color in the IR camera. Therefore, we conclude that the IR modulation mechanism of BLEE is similar to the brightness change mechanism of chameleon skin.

To clarify the similarity between the brightness change mechanism of chameleon skin and the IR modulation mechanism of BLEE, we replaced Figures 1 (A and B) with Figure R1, and we also modified the bio-inspired descriptions in the manuscript, in Page 3, Paragraph 2.

“...The brightness change of chameleon skin involves tuning the interaction between sunlight and iridophores by moving the melanosomes, which would not change the wavelength of reflected light but change the intensity of reflected light. It is similar to the working principle of an IR camera that uses fake color to represent the light intensity of IR ranging from 7.5 to 14 μm . Due to the tunable IR reflectance of the IR modulating system, different IR signals could be regulated and result in an adaptable IR image. Therefore, the brightness change mechanism of a chameleon inspires the design of a reversible liquid metal-based IR-modulating system.”

Figure R1. (A) Schematics of the mechanism of altering the brightness of a chameleon’s skin through the expansion/shrink of melanophores in the superficial iridophores of chameleons. (B) Schematic illustration of BLEE film modulating the IR reflectance by applying mechanical action.

Comment 2:

“Dynamic IR regulation of BLEE is studied in the range of 0~1,500 % of strain, and this number is considerably large. Although strain-dependent IR regulation is an interesting concept that stems from biological systems, there is no need to follow the exact anatomies of the color-changing animals. In this regard, areal strain, for such high value, might not be practical for dynamic systems. In other similar studies, the required strain was considerably smaller, e.g. *Science*. 359, 1495-1500 (2018), *ACS Nano*. 15, 17299-17309 (2021), and it was capable to prepare electrically actuated camouflage devices for practical modulation. Recent study on IR regulator also demonstrated electrically controlled device. [*Nat. Commun.* 14, 5087 (2023)]”

Response: We thank the reviewer for the comment. In the references mentioned by the referees ([*Science*. 359, 1495-1500 (2018)]; [*ACS Nano*. 15, 17299-17309 (2021)]), they used areal strain to modify the IR reflectance of materials, and the IR modulation mechanism of BLEE is different from these reported works. These referenced works used the wrinkles of polymer to tune the roughness of the surface and achieved specular reflectance-based IR camouflage. In our work, we utilized the shape-morphing property of liquid metal to provide *both specular reflectance- and total reflectance-based IR camouflage, which exhibits 61.2% specular reflectance change and 44.8% total reflectance change, respectively*, as shown in the following Table R1. The total reflectance change in our work performs the best. We select Ecoflex as the polymer matrix for the BLEE since it exhibits low Young’s modulus and large tensile limit. It can help to maximize the expansion of BLEE and show the limit of the optical performance of BLEE. Therefore, we studied the range of 0~1500% of strain in this work. Since the fabrication method of liquid metal- elastomer in our work is general, we further test the performance of liquid metal based-IR modulating system by using styrene ethylene butylene styrene (SEBS) polymer matrix with Young’s modulus higher than Ecoflex. As shown in Figure R2 and Table R1, BLEE made from the SEBS matrix achieved higher total reflectance change (27.6%) under lower areal strain (137%) than the results published in [*Science*. 359, 1495-1500 (2018)].

Science, 359, 1495-1500 (2018)	Reflectance	Areal strain (0%)	Areal strain (230%)	Reflectance change
	Total	71±3%	96±1%	25%
	Specular	23±1%	88±3%	65%
ACS Nano, 15, 17299-17309 (2021)	Reflectance	Areal strain (-50%)	Areal strain (0%)	Reflectance change
	Total	32.2±2.6%	58.2±5.2%	26%
	Specular	7.9±0.8%	36.4±1.6%	29%
	Reflectance	Areal strain (0%)	Areal strain (50%)	Reflectance change
	Total	58.2±5.2%	41.2±2.3%	17%
Our work (Ecoflex)	Reflectance	Areal strain (0%)	Areal strain (1500%)	Reflectance change
	Total	29.8±1.8%	74.6±1.6%	44.8%
	Specular	9.6±1.4%	70.8±2.8%	61.2%
Our work (SEBS)	Reflectance	Areal strain (0%)	Areal strain (137%)	Reflectance change
	Total	18.8±0.9%	46.4±2.3%	27.6%
Our work (SEBS)	Reflectance	Areal strain (0%)	Areal strain (525%)	Reflectance change
	Total	18.8±1.6%	61.7±1.3%	42.9%

Table R1. Summary of the total reflectance and specular reflectance of the IR modulating materials in previous works and our work.

Figure R2. The reflectance, transmittance, and absorptance spectra of BLEE film made from the SEBS matrix after applying different areal strains (0%, 137%, 525%).

This work [*Nat. Commun.* 14, 5087 (2023)] regulated the IR emissivity by modifying the carrier concentration in the surface depletion layer of aluminum-doped zinc oxide nanocrystals. The developed regulator achieved an IR regulation of 41% within the long-wave infrared (LWIR:7.5–13 μm) atmospheric windows. This reported dynamic IR emissivity regulator is not flexible and soft, therefore it is unable to tightly cloak complex three-dimensional structures. Besides, the implementation of this dynamic IR emissivity regulator depends on the precise packaging of the electrode with electrolyte, which restricts its patterning. Compared to this recent work, BLEE film exhibits some unique advantages, for example, it doesn't require injecting electrolyte and doesn't need complex circuit design. BLEE film performs good mechanical properties and can match the target object well for IR camouflage. Moreover, BLEE film can achieve complex patterns by simply cutting and masking.

To compare with recent works that focus on the electrically actuated IR regulator, we added the following statement in the Main Text, Page 10, Paragraph 1, and we also cited [*Science*. 359, 1495-1500 (2018)], [*ACS Nano*. 15, 17299-17309 (2021)], [*Nat. Commun.* 14, 5087 (2023)] as ref. [6], ref. [17], ref. [3], respectively.

"Our proposed method of liquid metal-based IR-modulating materials is not only applicable in Ecoflex but also can be expanded to other polymer matrix, such as styrene ethylene butylene styrene (SEBS), and polydimethylsiloxane (PDMS) (figs. S16, S17 and table S4). By using the SEBS polymer matrix with Young's modulus higher than Ecoflex, BLEE achieved higher total reflectance change (27.5%) under lower areal strain (137%) than the Ecoflex matrix, which proved BLEE could be potentially driven by electrical devices for practical modulation^{6,17}."

Comment 3:

"The internal structure of BLEE remains unclear: Figure 1B illustrates that EGaIn exists in the form of discrete droplets inside Ecoflex matrix and so they are after applying areal strain, but it is presented that Ecoflex and EGaIn are mechanically mixed, according to Materials and methods section. Since the ratio of Ecoflex/EGaIn can be as high as 1:10, it is anticipated that EGaIn will create a quasi-continuous layer within the Ecoflex matrix, which will lead to the discussion that illustration in Figure 1B can be misleading. Although many SEM and optical images of BLEE are presented in Figures and Supplementary Information, more detailed structure from topological perspective should be clarified. Cross-section images also should be more informative. These investigations will help the authors to conduct more in-depth theoretical analysis on the current system, which is relatively weak in the current version."

Response: We thank the reviewer for the comment. To clarify the internal structure of BLEE films, we further examined the cross-section scanning electron microscope (SEM) of Ecoflex/EGaIn=1:2, 1:4, 1:6, 1:8, and 1:10 in Figures R3 and R4. From the SEM analysis, we observed the bilayered structure, where the pure Ecoflex layer was under the red dashed line

and the Ecoflex/EGaIn layer was above the red dashed line. In the cross-section SEM images with higher magnification, liquid metal droplets cannot be observed in the samples with lower liquid metal concentrations (1:2 and 1:4) (Figure R4). With higher liquid metal concentrations (1:6, 1:8, and 1:10), liquid metal droplets (blue dashed circles in Figure R4) at the cross-section can be easily found. The droplets seem to be closely packed with each other.

Figure R3. SEM images of the front, back, and cross-section of BLEE film with Ecoflex/EGaIn mass ratio of 1:10.

Figure R4. The SEM images under different magnifications (600 \times and 1200 \times) of BLEE films prepared by different mass ratios of Ecoflex/EGaIn (1:2, 1:4, 1:6, 1:8, and 1:10).

We further used X-ray microscope (XRM) to provide three-dimensional information on the deformation process of liquid metal droplets within the elastomer in Figures R5, R6, and Movie S1-S4. From the three-dimensional reconstructor analysis, the liquid metal droplets are randomly dispersed in the elastomer without areal strain. There are more liquid metal droplets closer packed in the composite with a mass ratio of 1:10 than in the composite with a mass ratio of 1:2. After applying an areal strain of 1500%, the liquid metal droplets transform into flakes. The gaps between liquid metal droplets in the composite with a mass ratio of 1:2 become larger due to the low concentration of liquid metal droplets in the composite. In contrast, high concentration of liquid metals in elastomer (mass ratio of 1:10) create a quasi-continuous layer due to the overlap of liquid metal flakes. In Figure R6, the xy, xz, and yz cross-sections of samples are presented to further support the above results.

As suggested by the reviewer, we have replaced Figure 1 in the manuscript with Figure R7. We modified the schematics of liquid metal droplets in Figures 1 (B and E). We added the cross-section SEM results (Figure R3) and XRM results (Figure R5) of samples with mass ratio of 1:10 in Figures 1 (D and E), respectively. We added the other cross-section SEM results (Figure R4) and XRM results (Figure R6) in the supplementary information. We also included Movie S1-S4 as supplementary materials.

We added the following statement in the Main Text, Page 8, Paragraph 1.

“From the XRM results in Fig. 1E, fig. S6, movies S1, S2, S3, and S4, the 3D reconstructor analysis shows that the liquid metal droplets are randomly dispersed in the elastomer. After applying areal strain of 1500%, the liquid metal droplets in the composite transform into flakes. High concentration of liquid metals in elastomer (mass ratio of 1:10) create a quasi-continuous layer due to the overlap of liquid metal flakes.”

Figure R5. The three-dimensional XRM results of BLEE films with a mass ratio of 1:10 before and after applying areal strain (1500%).

Figure R6. (A) The three-dimensional XRM results and (B) the two-dimensional cross-section results of BLEE films with mass ratios 1:2 and 1:10 before and after applying areal strain (1500%). For the stretched BLEE films, the position of the y axle in the xy cross-section and the z axle in the xz cross-section coincide with the sample plane.

Figure R7. Concept and bio-inspired design of bilayered Ecoflex/EGaIn film (BLEE) enabling IR modulation. (A) Schematics of the mechanism of altering the brightness of a chameleon's skin through the expansion/shrink of melanophores in the superficial iridophores of chameleons. (B) Schematic illustration of BLEE film modulating the IR reflectance by applying mechanical action. (C) Schematic illustration of the fabrication process of BLEE film. (D) SEM images of the front, back, and cross-section of BLEE film with Ecoflex/EGaIn mass ratio of 1:10. (E) Optical appearance of BLEE film with Ecoflex/EGaIn mass ratio of 1:10 before and after applying areal strain of 1500%, images from optical microscopy and X-ray microscope of the morphology change of EGaIn droplets before and after stretching.

Comment 4:

“Mechanical properties of BLEE films at different mass ratios are shown in Figure 2G, and it shows that the modulus of BLEE at high EGaIn portion is higher. Why?”

Response: We thank the reviewer for the comment. As shown in Figure R8 and Table R2, when a small amount of liquid metal was added into Ecoflex, Young’s modulus of composite with Ecoflex/EGaIn mass ratio of 1:2 was lower than pure polymer. However, with the increasing concentration of liquid metal droplets, the Young’s moduli of composites increase. We reason that liquid metal without shear resistance can soften the Ecoflex matrix, but the oxide layer formed between liquid metal droplets and polymer can stiffen the composite [*Advanced Functional Materials* 31(1), 2005804 (2021)]. By examining the SEM images of liquid metal droplets taken from liquid metal-based elastomer composites with different mass ratios in Figure R9, the sizes of liquid metal droplets decrease with the increasing concentration of liquid metal in the composite. It is attributed that the viscosity of the mixture increases rapidly with adding a larger amount of liquid metal during the preparation process. The increased viscosity promotes shear stress of the mixture, which leads to the breakup of the liquid metal droplets, smaller average size distributions, and generation of more gallium oxide layer [*Soft Matter* 17(36), 8269-8275 (2021)]; [*Soft Matter* 16(28), 6608-6618 (2020)]. Therefore, it stiffens the composite with a high concentration of liquid metal and results in a higher Young’s modulus.

We added the mechanical properties of sample 1:0 (pure Ecoflex) in our work and replaced Figures 2G and S12 with Figures R8 (A and B), respectively. We added Figure R9 and Table R2 as Figure S1 and Table S3 in the supplementary information, respectively. To clarify the explanation of the mechanical properties of BLEE, we also added the following statement in the Main Text, Page 10, Paragraph 1 to further explain this phenomenon.

“In general, when the liquid metal inclusions without shear resistance replace the solid elastic polymer, the addition of liquid metal droplets will soften the composite. However, when the strong interfacial effects occur between the solids and liquid, such as the presence of a stiff oxide layer that forms at the liquid metal/Ecoflex interface (fig. S13), the composite will be effectively stiffened⁴³. As shown in Fig. 2G, fig. S12, and table S3, the Young’s moduli of composites with lower liquid metal concentrations (Ecoflex/EGaIn mass ratio of 1:2, 1:4, 1:6) in Ecoflex are smaller than the pure Ecoflex. However, with the increase of liquid metal concentrations, Young’s modulus gradually increases because the presence of the oxide layer inhibits the deformation of liquid metal droplets and stiffens the composites⁴³. When the mass ratios increase to 1:8 and 1:10, their Young’s moduli are even higher than that of pure Ecoflex.”

Figure R8. (A) The measured stress-strain curves and (B) The linear fitting stress-strain curves of BLEE films with different mass ratios of Ecoflex/EGaIn (1:0, 1:2, 1:4, 1:6, 1:8, and 1:10).

Samples	1:0	1:2	1:4	1:6	1:8	1:10
Young's Modulus (MPa)	0.102	0.060	0.077	0.088	0.138	0.167

Table R2. Young's moduli of BLEE films with different mass ratios of Ecoflex/EGaIn (1:0, 1:2, 1:4, 1:6, 1:8, and 1:10) are calculated from Figure R8.

Figure R9. (A) The SEM images and (B) size distributions of liquid metal droplets extracted from liquid metal-based elastomer composites with different mass ratios of Ecoflex/EGaIn (1:2, 1:4, 1:6, 1:8, and 1:10).

Comment 5:

"In Figure 5, it is mentioned that EGaIn-based programmable IR encryption film is realized in 3D structure, but this expression is not very convincing: by adding a third layer, now the device is in trilayer configuration, but increase in thickness in out-of-plane direction is almost insignificant. It is shown that overlapped information can be distinguished by adding the third layer, yet I would like to recommend the authors to find some different term."

Response: We thank the reviewer for the comment. As suggested by the reviewer, we changed "3D structure" in the manuscript and supplementary information to "multi-layered structure".

We also modified the title:

From "Chameleon-Inspired Tunable Three-Dimensional Infrared-Modulating System via Stretchable Liquid Metal Microdroplets in Elastomer Film"

To "Chameleon-Inspired Tunable Multi-layered Infrared-Modulating System via Stretchable Liquid Metal Microdroplets in Elastomer Film".

Comment 6:

"Please check again for minor mistakes, e.g. "high-reflectacne state"; in Figure 1E, etc."

Response: We apologize for the spelling error, and we replaced Figure 1E with Figure R10.

Figure R10. Schematic illustration of BLEE film modulating the IR reflectance by applying mechanical action.

Comments from Reviewer #2

“In this work, Zhang et al. presented liquid metal-based infrared-modulating materials and systems with multiple modes to regulate the infrared reflection. This work seems interesting but the innovation needs to be improved. Therefore, I recommended it to be published after some revisions with the following comments:”

Response: Thank the reviewer for the encouraging comments that help us to improve the quality of our manuscript. We have revised our manuscript to address the concerns of the reviewer.

Comment 1:

“This material must be stretched to change the infrared reflectance, so what is its specific application scenario?”

Response: We thank the reviewer for the comment. In this work, we have demonstrated that our flexible bilayered/multi-layered Ecoflex/EGaIn infrared (IR) modulating materials have potential in IR camouflage, encoding/decoding, painting/writing, and encryption. Stretchable IR regulators can be also used in the field of personal thermal management [*Nature Communications* 10, 1947 (2019)], radiative heat management [*Nano Letters* 21, 9, 4106–4114 (2021)], and finger motion sensing [*Materials Today* 45, 44-53 (2021)].

Besides that, there are also some published works on strain-induced dynamic color change in visible light, such as structurally colored materials [*Nature Materials* 21, 1014–1018 (2022)], donor-acceptor Stenhouse adducts [*Nature Chemistry* 15, 332–338 (2023)], fluorescence [*Journal of the American Chemical Society* 145, 49, 26799–26809 (2023)], and liquid crystalline elastomer [*Advanced Science* 9(36), 2205325 (2022)]. These materials have been demonstrated in the applications of mechanosensitive healthcare materials, strain, and stress sensing for human-computer interaction, optical encryption/decryption, and wide color gamut displays. Thus, the mechanically sensitive IR-modulating materials can be an effective candidate for the above applications in dark environments.

To clarify the application scenario, we provided an additional statement on potential applications of stretchable IR regulators in the Main Text, Page 3, Paragraph 1.

“...Specifically, mechanical stimulated IR regulators can be also used in the fields of personal thermal management⁹, radiative heat management¹¹, finger motion sensing¹², and etc...”

Comment 2:

“A similar principle of changing the reflectance has been reported. What are the advantages of your work? (*Materials Today Chemistry* 24 (2022) 100911)”

Response: We thank the reviewer for the comment. Both this work [*Materials Today Chemistry* 24, 100911 (2022)] and ours show a strain-dependent IR reflectance change. Compared to this work and other metal based-IR modulating systems, our liquid metal-based elastomer exhibits some unique advantages, as shown in Table R3.

	Materials	Mechanization	IR reflectance change (with the increase of areal strain)	Mass ratio-dependent IR reflectance	Temperature-dependent IR reflectance	Use the evaporated metal film
Materials Today Chemistry 24, 100911 (2022)	Au-SEBS membrane	Microcracks at the metal layer	97%-37%	No	No	Yes
Our work	LM-Ecoflex film	Deformation of LM droplet	29.8±1.8%-74.6±1.6%	Yes	Yes	Yes

Table R3. Summary of the comparison between our work and other metal based-IR modulating systems.

Liquid metal-based IR-modulating materials show mass ratio-dependent IR reflectance. The IR reflectance of our materials can be changed by adjusting the mass ratio of Ecoflex/EGaIn, as shown in Figure 2E, the intensity of IR reflectance increases with the increasing concentration of EGaIn.

The evaporated metal film-based IR-modulating materials (*Materials Today Chemistry*, 2022, 24, 100911) employ the metals with very high melting points. On the contrary, in our research, we proposed a general method to develop low melting point alloy-based IR-modulating materials, which regulate the IR reflection by controlling the deformation of liquid metal droplets in the polymer matrix. By utilizing liquid metals with different melting points, we can programmably control the phase change of liquid metal particles/droplets within the elastomer. For example, by selecting alloys with different melting points, we achieved a temperature-dependent IR display, as shown in Figure 4I.

Compared with evaporated metal film-based IR-modulating materials, the intensity of IR reflectance from our liquid metal-based elastomer becomes stronger with the increasing areal strain. We have developed a multi-layered programmable encryption system by evaporating a specific gold film pattern on our liquid metal-based elastomer film, where various and more complex IR patterns can be generated and switched under visible or IR mode.

To clarify the advantages of our developed liquid metal-based elastomer composite, we modified and added the following statement in the Main Text, Page 18, Paragraph 2, and we also cite this article as ref. [44] in our manuscript.

“High mp metals including Au, Al and copper (Cu) are usually evaporated on polymer film and widely used as IR-modulating materials^{6,9,17,44}. Compared with the evaporated metal film-based IR-modulating materials, our BLEE film exhibits unique advantages such as liquid metal proportion depended IR reflectance, programmable IR display enabled by different low

mp alloys, and multilayer structural design by combining BLEE film with evaporated metal film together to achieve more complex IR patterns.”

Comments from Reviewer #3

“The manuscript presents a layered structure consisting of liquid metal droplets in silicone elastomer (Ecoflex), that allows IR modulation through IR reflectance. The deformation of liquid metal microdroplets enables the IR reflectance to change from a relatively low state to a high reflectance state, which is regulated under mechanical strain. This is a proof-of-concept study showing the potential utility of EGaIn/Ecoflex composites as IR camouflage; however, the presented results are not sufficient to support the major claims of the study. Thus, with major revisions this work would be better suited for a more specialized journal.”

Response: We thank the reviewer for the supportive comments that helped improve the quality of the manuscript. We have revised our manuscript and carried out additional experiments to address the concerns of the reviewer.

Comment 1:

“The primary concern in the present manuscript revolves around the insufficient characterization of the liquid metal droplet structure within the composite. While SEM analysis was conducted, it lacks detailed information about the liquid metal (LM) droplets, such as their size when embedded in the composite.”

Response: We thank the reviewer for the comment. The liquid metal droplets are embedded in the composite, and they cannot be simply observed by the scanning electron microscope (SEM) analysis of the polymer composite. Therefore, we develop a method to extract the mixed liquid metal droplets in the Ecoflex precursor to conduct SEM characterization. To obtain the liquid metal droplets within Ecoflex, Ecoflex precursor was first mechanically mixed with EGaIn with different mass ratios (1:2, 1:4, 1:6, 1:8, and 1:10) at 700 rpm for 5 minutes, which is the same as the fabrication process of bilayered Ecoflex/EGaIn film (BLEE). After completely mixing, 5 ml hexane was added to the mixture and stirred at 300 rpm for 10 minutes. The Ecoflex precursor gradually dissolved in the hexane solvent. The mixture was centrifugated at 2000 rpm for 10 minutes, and the sediment was redistributed in hexane. After repeating the above purification steps twice, the Ecoflex precursor was completely removed. Finally, the liquid metal droplets dispersed in the hexane were dropped on the clean silicon wafer to prepare the samples for SEM analysis.

The SEM images and the size distributions of liquid metal droplets from composites with different mass ratios are shown in Figure R11. The size distribution of liquid metal droplets ranges from a few micrometers to tens of micrometers. As for the liquid metal microdroplets mix with polymer at mass ratio of 1:2, the average diameter is 17.94 μm . By increasing the liquid metal proportion in the composites, the size of individual microdroplets decreases. Liquid metal microdroplets in polymer with mass ratio of 1:10 exhibit the smallest average size of 6.68 μm . It is attributed that the viscosity of the mixture increases with the increasing amount of liquid metal in the composite. The enhanced shear stress during mixing leads to the breakup of the liquid metal droplets and smaller average size distributions [*Soft Matter* 17(36), 8269-8275 (2021)]; [*Soft Matter* 16(28), 6608-6618 (2020)].

We added Figure R11 as Figure S1 in the supplementary information and added the following statement in the Main Text, Page 7, Paragraph 2.

“The SEM images and the size distributions of liquid metal droplets from composites with different mass ratios are shown in fig. S1. The size distribution of liquid metal droplets ranges from a few micrometers to tens of micrometers. As for the liquid metal microdroplets mix with polymer at mass ratio of 1:2, the average diameter is 17.94 μm . By increasing the liquid metal concentration in the composites, the size of individual microdroplets decreases. Mixing liquid metal microdroplets with polymer at mass ratio of 1:10 results in the smallest average size of 6.68 μm . It is attributed that the viscosity of the mixture increases with the increasing

amount of liquid metal in the composite. The enhanced shear stress during mixing leads to the breakup of the liquid metal droplets and smaller average size distributions^{38,39}.”

We also added the extraction process of liquid metal droplets from composites in the supplementary information.

“To obtain the liquid metal droplets within Ecoflex, Ecoflex precursor was first mechanically mixed with EGaIn at different mass ratios (1:2, 1:4, 1:6, 1:8, and 1:10) at 700 rpm for 5 minutes, which is the same as the fabrication process of BLEE. After mixing, 5 ml hexane was added to the mixture and stirred at 300 rpm for 10 minutes. The Ecoflex precursor gradually dissolved in the hexane solvent. The mixture was centrifugated at 2000 rpm for 10 minutes, and the sediment was redistributed in hexane. After repeating the above purification steps twice, the Ecoflex was completely removed. Finally, the liquid metal particles dispersed in the hexane were dropped on the cleaned silicon wafer to prepare the samples for SEM analysis.”

Figure R11. (A) The SEM images and (B) size distributions of liquid metal droplets extracted from composites with different mass ratios of Ecoflex/EGaIn (1:2, 1:4, 1:6, 1:8, and 1:10).

Comment 2:

“Moreover, it is essential to investigate whether there are any size variations in LM droplets at different EGaIn/Ecoflex ratios.”

Response: We thank the reviewer for the comment. According to Figure R11 in Comment 1, the sizes of liquid metal droplets decrease with the increasing amount of liquid metal proportion in composites. It is attributed that the viscosity of the mixture increases with the increasing amount of liquid metal in the composite [*Soft Matter* 16(28), 6608-6618 (2020)]. The enhanced shear stress during mixing leads to the breakup of the liquid metal droplets and smaller average size distributions [*Soft Matter* 17(36), 8269-8275 (2021)].

Comment 3:

“The major claim of the paper hinges on the stretching of the composite, resulting in the deformation or shape change of the EGaIn droplets within the elastomer. Unfortunately, this shape change has not been adequately characterized in the manuscript, thereby leaving a gap in the support for the paper's major claim.”

Response: We thank the reviewer for the helpful comment. The optical microscope images in supplementary information Figure S3 show the deformation of liquid metal droplets under different areal strains. As suggested by the reviewer, we further used an X-ray microscope (XRM) to provide three-dimensional information on the deformation process of liquid metal droplets within the elastomer, as shown in Figures R12, R13, and Movie S1-S4. From the three-dimensional reconstructor analysis shown in Figures R12 and R13, the liquid metal droplets are randomly dispersed in the elastomer without areal strain. There are more liquid metal droplets closer packed in the composite with mass ratio of 1:10 than in the composite with mass ratio of 1:2. After applying areal strain of 1500%, the liquid metal droplets

transform into flakes. The gaps between liquid metal droplets in the composite with mass ratio of 1:2 become larger due to the low concentration of liquid metal droplets in the composite. In contrast, high concentration of liquid metals in elastomer (mass ratio of 1:10) create a quasi-continuous layer due to the overlap of liquid metal flakes. In Figure R13, the xy, xz, and yz cross-sections of samples are presented to further support the above results. The cross-section results from XRM can clearly show the shape and distribution of liquid metal droplets during the shape-changing process.

We added the following statement in the Main Text, Page 8, Paragraph 1.

“From the XRM results in Fig. 1E, fig. S6, movies S1, S2, S3, and S4, the 3D reconstructor analysis shows that the liquid metal droplets are randomly dispersed in the elastomer. After applying areal strain of 1500%, the liquid metal droplets in the composite transform into flakes. High concentration of liquid metals in elastomer (mass ratio of 1:10) create a quasi-continuous layer due to the overlap of liquid metal flakes.”

To clearly show the deformation of liquid metal droplets within the elastomer under areal strain, we added Figure R12 in Figure 1E and added Figure R13 as Figure S6 in the supplementary information. We further added Movie S1-S4 as supplementary material to show the three-dimensional liquid metal droplets and internal slice images.

Figure R12. The three-dimensional XRM results of BLEE films with a mass ratio of 1:10 before and after applying areal strain (1500%).

Figure R13. (A) The three-dimensional XRM results and (B) the two-dimensional cross-section results of BLEE films with mass ratios 1:2 and 1:10 before and after applying areal strain (1500%). For the stretched BLEE films, the position of the y axle in the xy cross-section and the z axle in the xz cross-section coincide with the sample plane.

Comment 4:

“Additionally, there is a lack of clarity regarding the relationship between the size of EGaIn droplets and the regulation of IR-reflectance. It would be beneficial to elucidate the specific size or size range at which IR-reflectance becomes more significant in the context of the study.”

Response: We thank the reviewer for the helpful comment. To clarify the influence of the size of liquid metal droplets within Ecoflex on infrared (IR) reflectance, we fixed the Ecoflex/EGaIn mass ratio of 1:6 and prepared 4 samples by different methods.

Sample #1: *mix 5 mins*: we mixed liquid metal and Ecoflex for 5 mins and prepared the BLEE with an average liquid metal droplet size of 12.43 μm (Figure R14 (i)), which followed the same method in the manuscript.

Sample #2: *mix 20 mins*: we extended the mechanical mixing time of liquid metal and Ecoflex from 5 mins in the manuscript to 20 mins, and prepared the BLEE with an average liquid metal droplet size of 3.22 μm (Figure R14 (ii)), which is smaller than the sizes in manuscript (as shown in Figure R14 (i)).

Sample #3: *probe sonication/sediment*: we further used probe sonication to prepare liquid metal nanoparticles. The liquid metal was sonicated for 15 mins in ethanol with a power of 300 W. The above solution was centrifuged at 400 rpm for 10 minutes. The sediment was collected and used to prepared BLEE film with an average liquid metal droplet size of 1.35 μm (Figure R14 (iii)).

Sample #4: *probe sonication/suspension*: the above suspension was further centrifugated. The particles with an average liquid metal droplet size of 0.63 μm were collected and used to prepare BLEE film (Figure R14 (iv)).

Note: The above as-prepared liquid metal droplets from either probe sonication or centrifugation were further transferred into hexane solution, and purified twice with 2000 rpm for 10 mins under centrifugation before being used to prepare BLEE films.

We measured the IR spectra of BLEE film prepared from the above 4 types of liquid metal droplets, the results are shown in Figure R15. From the specular reflectance results, we observed that BLEE films prepared by “Sample #1: *mix 5 mins*”, “Sample #2: *mix 20 mins*” and “Sample #3: *probe sonication/sediment*” showed IR reflectance changes with different areal strains. However, the IR reflectance spectra of BLEE film prepared by “Sample #4: *probe sonication/suspension*” did not show obvious change under areal strain.

Based on the above experimental results, we can conclude that the size of liquid metal microdroplets plays an important role in liquid metal-based IR-modulating materials. The IR reflectance decreases with the reduced size of liquid metal droplets. BLEE film prepared from liquid metal nanodroplets cannot deform, and lose the capability of IR regulation. According to the Young-Laplace equation, the external force should be larger than $2\gamma/R$ for triggering the liquid metal particle deform, in which γ is the surface tension of liquid metal (0.624 N/m) and R is the radius of the liquid metal particle. As for the liquid metal nanodroplets, the external trigger force becomes very high so the deformation of liquid metal nanodroplets becomes difficult [RSC advances 8(29), 16232-16242 (2018)].

We further used COMSOL Multiphysics to simulate the deformation of liquid metal droplets with different particle sizes within Ecoflex matrix when applying 1500% areal strain (maximum areal strain in our work). We defined the deformation of liquid metal droplets as $(R' - R)/R$. R was the radius of liquid metal droplets without areal strain and R' was the radius of liquid metal droplets with areal strain of 1500% as illustrated in Figure R16A. Young’s modulus of Ecoflex matrix (E_0) was set to 0.102 MPa as measured in Table R5. Poisson’s ratio of Ecoflex matrix (ν_0) was set to 0.4287 [IEEE/ASME Transactions on Mechatronics 18(5), 1602-1611 (2012)]. Effective Young’s modulus (E_{eff}) of liquid metal droplet was calculated with the following equation [Advanced Functional Materials 31(1), 2005804 (2021)].

$$E_{eff} = \frac{E_i t}{2R} \left(\frac{5E_i t + 2E_0 R(7 - 5\nu_i)}{E_i t(7 + 5\nu_i) + 10E_0 R(1 - \nu_i^2)} \right) \quad (R1)$$

where the Young’s modulus of gallium oxide E_i was set to be 30 GPa [Particle & Particle Systems Characterization 38(10), 2100141 (2021)], Poisson’s ratio of gallium oxide (ν_i) was set to 0.3 [Langmuir 34(1), 234-240 (2018)], and the thickness of the gallium oxide layer (t) was 1.4 nm according to the transmission electron microscope (TEM) results in Figure R17. We simulated the deformation of liquid metal droplets with diameters from 0.1 μm to 40 μm under applied areal strain of 1500%, and the theoretical calculation results were shown in Figures R16 (B and C). From the calculation results, we find that the deformation of liquid

metal droplets reduces with the decreasing size of liquid metal droplets. Especially, when the size of droplet is below 3.22 μm , the deformation reduces rapidly with the decreasing size. The liquid metal droplet with size of 0.1 μm hardly deform under areal strain. Such size-dependent deformation of liquid metal droplets has been also observed in liquid metal-based flexible circuits [*Science* 378(6620), 637-641 (2022)].

To clarify the influence of the size of liquid metals on the IR reflectance, we added Figures R14, R15 and R16 as Figures S9, S10 and S11, respectively, in the supplementary information. We further illustrated the size limitation of liquid metal droplets for IR regulation in the Main Text, Page 9, Paragraph 2.

“Well-designed droplet size of liquid metal is also important in the BLEE-based IR-modulating materials. We fabricated BLEE films with different liquid metal droplet size distributions and measured their IR reflectance in figs. S9 and S10. BLEE films composed of hundred-nanometer liquid metal droplets could not show mechanically stimulated IR modulation, since the liquid metal nanodroplets cannot undergo deformation in the polymer matrix⁴¹. We also performed theoretical simulations and confirmed this size-dependent deformation of liquid metal droplets in Ecoflex matrix (see the Note in supplementary information and fig. S11), which has been also observed in previous work⁴².”

We also added a Note to elucidate the reason in detail in the supplementary information.

“Note 1: To clarify the influence of the size of liquid metal droplets within Ecoflex on IR reflectance, we fixed the Ecoflex/EGaIn mass ratio of 1:6 and prepared 4 samples by different methods.

Sample #1: mix 5 mins: we mixed liquid metal and Ecoflex for 5 mins and prepared the BLEE with an average liquid metal droplet size of 12.43 μm (fig. S9 (i)), which followed the same method in the manuscript.

Sample #2: mix 20 mins: we extended the mechanical mixing time of liquid metal and Ecoflex from 5 mins in the manuscript to 20 mins, and prepared the BLEE with an average liquid metal droplet size of 3.22 μm (fig. S9 (ii)), which is smaller than the sizes in manuscript (as shown in fig. S9 (i)).

Sample #3: probe sonication/sediment: we further used probe sonication to prepare liquid metal nanoparticles. The liquid metal was sonicated for 15 mins in ethanol with a power of 300 W. The above solution was centrifuged at 400 rpm for 10 minutes. The sediment was collected and used to prepared BLEE film with an average liquid metal droplet size of 1.35 μm (fig. S9 (iii)).

Sample #4: probe sonication/suspension: the above suspension was further centrifugated. The particles with an average liquid metal droplet size of 0.63 μm were collected and used to prepare BLEE film (fig. S9 (iv)).

Note 2: The above as-prepared liquid metal droplets from either probe sonication or centrifugation were further transferred into hexane solution, and purified twice with 2000 rpm under centrifugation for 10 mins before being used to prepare BLEE films.”

“From the SEM and IR specular reflectance analysis (figs. S9 and S10), we observed that BLEE films prepared by “Sample #1: mix 5 mins”, “Sample #2: mix 20 mins” and “Sample #3: probe sonication/sediment” showed IR reflectance changes with different areal strains. However, the IR reflectance spectra of BLEE film prepared by “Sample #4: probe sonication/suspension” did not show obvious change under areal strain. It is attributed that the applied external force should be larger than $2\gamma/R$ for triggering the liquid metal particle deform according to the Young-Laplace equation, in which γ is the surface tension of liquid metal (0.624 N/m) and R is the radius of the liquid metal particle. As for the liquid metal nanodroplets, the applied external force becomes very high so the deformation of liquid metal nanodroplets becomes difficult⁷.”

“We used COMSOL Multiphysics to simulate the deformation of liquid metal droplets with different droplet sizes within Ecoflex matrix when applying 1500% areal strain. We defined the deformation of liquid metal droplets as $(R' - R)/R$. R was the radius of liquid metal droplets without areal strain and R' was the radius of liquid metal droplets with areal strain of 1500% as illustrated in fig. S11A. Young’s modulus of Ecoflex matrix (E_0) was set to 0.102 MPa as measured in table S3. Poisson’s ratio of Ecoflex matrix (ν_0) was set to 0.4287³. Effective Young’s modulus (E_{eff}) of liquid metal droplet was calculated by Equation (S9)⁴:

$$E_{eff} = \frac{E_i t}{2R} \left(\frac{5E_i t + 2E_0 R(7-5V_i)}{E_i t(7+5V_i) + 10E_0 R(1-V_i^2)} \right) \quad (S9)$$

where the Young's modulus of gallium oxide E_i was set to be 30 GPa⁵. Poisson's ratio of gallium oxide (V_i) was set to 0.3⁶. The thickness of the gallium oxide layer (t) was 1.4 nm according to the transmission electron microscope (TEM) results in fig. S13. We simulated the deformation of liquid metal droplets with diameters from 0.1 μm to 40 μm under applied areal strain of 1500%, and the theoretical calculation results were shown in figs. S11 (B and C)."

Figure R14. (A) The SEM images and (B) size distributions of liquid metal droplets in BLEE films with a mass ratio of 1:6 prepared by different methods ((i) mix 5 mins, (ii) mix 20 mins, (iii) probe sonication/sediment, and (iv) probe sonication/suspension).

Figure R15. The reflectance, transmittance, and absorbance spectra of BLEE films prepared by different liquid metal droplet sizes after applying different areal strains (0%, 137%, 525%, and 1500%).

Figure R16. (A) Schematic illustration of liquid metal droplet with and without areal strain, (B) Calculated deformations of liquid metal droplets with different droplet sizes, (C) Simulation results of liquid metal droplets with sizes from 0.1 μm to 40 μm when under applied areal strain of 1500%.

Comment 5:

“Is there any surface oxidation observed in the liquid metal droplet? The manuscript lacks information on the surface properties of the LM droplets and how these properties might influence the IR reflectance of the composites. Providing insights into the surface characteristics of the liquid metal droplets and their potential impact on IR reflectance would contribute to a more comprehensive understanding of the study.”

Response: We thank the reviewer for the comment. According to the previous works, the liquid metal forms a thin oxide layer on its surface with thickness range from 0.7 nm to 3 nm [ACS applied materials & interfaces 6(21), 18369-18379 (2014)]; [Proceedings of the National Academy of Sciences 111(39), 14047-14051 (2014)]; [Angewandte Chemie International Edition 54(43), 12809-12813 (2015)]; [Micromachines 14(1), 17 (2022)].

To better analyze the surface oxidation of liquid metal droplets in our work, we extracted the liquid metal droplets from the composite by using the aforementioned method (Comment 1) and examined the thickness of the oxide layer with TEM and energy dispersive spectrometer (EDS) mapping (Figure R17). From the high-resolution TEM image in Figure R17B, we determined the thickness of the gallium oxide layer was about 1.4 nm, which is consistent with previous works. To clarify the influence of gallium oxide on the IR reflectance, we employed Beer-Lambert Law. It shows that

$$I = I_0 e^{-al} \quad (R2)$$

$$a = \frac{4\pi k}{\lambda} \quad (R3)$$

where I is the intensity of transmitted light, I_0 is the intensity of incident light, a is the absorption coefficient, l is the optical path length, k is the extinction coefficient, and λ is the wavelength of light. According to the equations, k and l are the two important factors influencing the intensity of transmitted light. In our study, l is the thickness of the oxide layer (t), which is 1.4 nm. Unfortunately, up to now, there have been no reports on k of gallium oxide. However, the strong peaks of Ga-O bending vibrations occur at 688 cm^{-1} ($14.5 \text{ }\mu\text{m}$) and 459 cm^{-1} ($21.7 \text{ }\mu\text{m}$), which are not in the detection range of the IR camera [Scientific Reports 12(1), 20181 (2022)]. Therefore, we assume that the k of gallium oxide should be relatively small in the range of 7.5 to $14 \text{ }\mu\text{m}$ [J. Chem. Eng. Data 42, 2, 342-345 (1997)]. Here, we use silica (SiO_2) and alumina (Al_2O_3) as examples to calculate the k value, and show how 1.4 nm thickness impact on the IR transmission. SiO_2 has the bands at 1187 cm^{-1} ($8.4 \text{ }\mu\text{m}$) and 1073 cm^{-1} ($9.3 \text{ }\mu\text{m}$) assigned to the asymmetric stretching O-Si-O and Si-O-Si modes, respectively, and the band at 451 cm^{-1} ($22.2 \text{ }\mu\text{m}$) assigned to the in-plane bending mode of Si-O bonds [Ceramics International 48(15), 22006-22017 (2022)]. Therefore, SiO_2 has a relatively large k , and has a peak of 1.985 at the wavelength of $9.328 \text{ }\mu\text{m}$ [Applied Optics 51, 6789-6798 (2012)]. According to the Beer-Lambert Law, we can calculate the transmittance (I/I_0) of SiO_2 with 1.4 nm thickness. It is 0.996 at the wavelength of $9.328 \text{ }\mu\text{m}$. Al_2O_3 has characteristic absorption peaks at 561 cm^{-1} ($17.8 \text{ }\mu\text{m}$), 753 cm^{-1} ($13.3 \text{ }\mu\text{m}$), and 818 cm^{-1} ($12.2 \text{ }\mu\text{m}$) [Polymer Bulletin 79(7), 5279-5303 (2022)]. Hence, Al_2O_3 has a peak of k (1.439) at the wavelength of $12.5 \text{ }\mu\text{m}$ [Applied Optics 51, 6789-6798 (2012)]. According to the Beer-Lambert Law, we can calculate the transmittance (I/I_0) of Al_2O_3 with 1.4 nm thickness. It is 0.998 at the wavelength of $9.328 \text{ }\mu\text{m}$. Therefore, we believe that gallium oxide with such a small thickness has negligible influence on IR transmission.

To clarify the thickness of the oxide layer on liquid metal droplets, we added Figure R17 as Figure S13 in the supplementary information.

Figure R17. (A) The TEM image, (B) high-resolution TEM image, and (C) EDS mapping images of liquid metal droplets.

Comment 6:

“Once more, the authors have presented characterizations of the film's surface roughness using optical imaging. However, it remains uncertain whether the tuning of specular reflectance was solely attributable to the deformation of the LM droplets, as there is a lack of supporting data. It is possible that the observed effect is linked to the formation and deformation of the LM droplets network within the elastomer matrix (*Science* 378,637-641(2022).DOI:10.1126/science.abo6631). Providing additional data or clarification on this aspect would strengthen the argument and enhance the overall credibility of the findings.”

Response: We thank the reviewer for the comment. We measured the surface roughness of our samples and explained the reasons in the supplementary information. In our work, we believe that the deformation of liquid metal droplets rather than the surface roughness results in the specular reflectance changes because of two main reasons. 1) Previous work [*ACS Nano* 15, 17299-17309 (2021)] tuned the specular reflectance by the surface roughness, and reported that the specular reflectance would increase with the decrease of the surface roughness. However, in our work, there is no defined relationship between the surface roughness and the specular reflectance. For example, the surface roughness of BLEE film with mass ratio of 1:4 is 0.916 μm , which is a bit lower than the sample with mass ratio of 1:6 (0.920 μm). However, the specular reflectance of sample 1:6 (66%) is much higher than sample 1:4 (46%) (Figure 3E). 2) Usually, a large surface roughness is needed to achieve the significant specular reflectance change in the previous work [*ACS Nano* 15, 17299-17309 (2021)]. For example, 29% increase in specular reflectance followed by 5.52 μm decrease of surface roughness. However, in our work, there is only 0.575 μm of surface roughness change (from pure Ecoflex to BLEE with mass ratio of 1:10), and it cannot achieve a 61.2% of specular reflectance change.

The work [*Science* 378,637-641(2022)] mentioned by the reviewer presented a liquid metal droplet network assembled with an acoustic field. This liquid metal droplet network was used as a flexible circuit, and performed stable electrical performance during large deformation. However, our liquid metal droplets embedded in Ecoflex do not form liquid metal droplet network before and after applying areal strain. We measured the resistance and electrical conductivity of BLEE film before and after applying areal strain, the results are shown in Table R4. The BLEE films without areal strain were insulating even at high liquid metal concentration, which is consistent with the results published in previous work [*Science* 378,637-641(2022)]. We observed that our BLEE film remained insulating after applying external strain. It demonstrated that there was no liquid metal network formed in BLEE films before and after applying the areal strain. Generally, liquid metal-based conductors require an additional step to sinter liquid metal droplets and achieve high electrical conductivity, which is absent in our work. Besides that, there was no liquid metal network observed in XRM results (Figures R12 and R13). It further proves that the deformation (from spheres to flakes) of liquid metal droplets instead of liquid metal network formation impacts on the IR modulating property of BLEE films.

Sample	Resistance ($\times 10^8 \Omega$)	Electrical conductivity ($\times 10^{-4} \text{ S/m}$)
1:2-0%	2.167 \pm 0.371	0.232 \pm 0.034
1:2-1500%	2.564 \pm 0.012	1.950 \pm 0.009
1:4-0%	2.661 \pm 0.059	0.800 \pm 0.016
1:4-1500%	2.661 \pm 0.052	1.504 \pm 0.030
1:6-0%	2.486 \pm 0.010	1.437 \pm 0.006
1:6-1500%	2.509 \pm 0.015	1.979 \pm 0.035
1:8-0%	2.396 \pm 0.076	1.071 \pm 0.035
1:8-1500%	2.564 \pm 0.095	1.507 \pm 0.062
1:10-0%	2.148 \pm 0.978	1.476 \pm 0.039
1:10-1500%	2.581 \pm 0.056	1.917 \pm 0.004

Table R4. Resistance and electrical conductivity of BLEE film before and after applying areal strain.

Comment 7:

“Furthermore, the rationale behind choosing Ecoflex over alternative elastomers, such as PDMS, remains unclear. It would enhance the comprehensibility of the study to include a comparison or discussion addressing the specific reasons for opting for Ecoflex and how it compares to other elastomeric materials.”

Response: We thank the reviewer for the comment. Ecoflex was used in this work since it exhibits excellent tensile performance, and can be utilized to test the IR performance for large liquid metal deformation. Our preparation method is general and can be expanded to other polymers.

To prove the generality of our method, we further prepared BLEE films with other polymer matrices. We selected polyurethane (PU), styrene ethylene butylene styrene (SEBS), and polydimethylsiloxane (PDMS) as typical polymer matrices for study. The mass ratio of Polymer/EGaIn was fixed at 1:6 for all samples prepared below. Firstly, we measured the mechanical properties of the sample with and without liquid metal, and the experimental results and their Young’s moduli are shown in Figure R18 and Table R5. As shown in Figure R18A, PU was broken up when the strain reached 250%, and PDMS and SEBS did not break up within the range of testing. Compared the Young’s moduli of the samples with and without adding liquid metal in Table R5, we observed that liquid metal softened all the composites. Ecoflex has the lowest Young’s modulus.

We further measured the IR specular reflectance of BLEE film prepared by PDMS and SEBS (Figure R19). Both of these two samples show the adjustable IR reflectance under areal strain. It demonstrated that liquid metal-based-IR modulating materials could be fabricated in other elastomer systems.

We added Figures R18 and R19 in the supplementary information as Figures S16 and S17, respectively. We also added the results of Young’s moduli (Table R5) as Table S4 in the supplementary information. Besides that, we added a statement in the Main Text, Page 10, Paragraph 1:

“Our proposed method of liquid metal-based IR-modulating materials is not only applicable in Ecoflex but also can be expanded to other polymer matrix, such as styrene ethylene butylene styrene (SEBS), and polydimethylsiloxane (PDMS) (figs. S16, S17 and table S4). By using the SEBS polymer matrix with Young’s modulus higher than Ecoflex, BLEE achieved higher total reflectance change (27.5%) under lower areal strain (137%) than the Ecoflex matrix, which proved BLEE could be potentially driven by electrical devices for practical modulation^{6,17}.”

And we added a Note in the supplementary information to elucidate the results.

“To prove the generality of our method of fabricating liquid metal-based IR modulating materials, the other three kinds of elastomer materials (polyurethane (PU), styrene ethylene butylene styrene (SEBS), and polydimethylsiloxane (PDMS)) have been selected. The mechanical testing results (fig. S16 and table S4) show that the addition of liquid metal softened the composite, and pure PU could not be further mixed with liquid metal due to its poor mechanical performance. The IR specular reflectance results show that BLEE films prepared by either SEBS or PDMS shows mechanical stimulated IR-modulating reflectance (fig. S17).”

Figure R18. (A) The measured stress-strain curves and (B) the linear fitting stress-strain curves of BLEE films made from different polymers with mass ratios of 1:0 and 1:6.

Samples	PU 1:0	SEBS 1:0	SEBS 1:6	PDMS 1:0	PDMS 1:6	Ecoflex 1:0	Ecoflex 1:6
Young's Modulus (MPa)	2.142	1.330	0.985	0.290	0.508	0.102	0.088

Table R5. Young's Moduli of different BLEE films prepared with different mass ratios of 1:0 and 1:6, which is calculated from Figure R18B.

Figure R19. The reflectance, transmittance, and absorbance spectra of BLEE films prepared by different polymer matrices after applying different areal strains (0%, 137%, 525% for SEBS, and 0%, 137%, 250% for PDMS).

Thanks again for your effort in reviewing the manuscript!

Yours truly,

Tao Deng
Professor, Zhi Yuan Chair
School of Materials Science and Engineering
Shanghai Jiao Tong University

REVIEWERS' COMMENTS

Reviewer #1 (Remarks to the Author):

Having carefully reviewed the manuscript, I am pleased to recommend its publication, as the revisions appear to have significantly strengthened the overall quality of the work.

Reviewer #2 (Remarks to the Author):

I have looked over the authors' point-by-point response and revisions. The authors have well addressed the comments. I would recommend its acceptance.

Reviewer #3 (Remarks to the Author):

Questions have been adequately addressed in the revised manuscript. I recommend publication with no change.